# Fast and slow feedforward inhibitory circuits for cortical odor processing

**Norimitsu Suzuki[1], Malinda LS Tantirigama[1,2†], K Phyu Aung[1], Helena HY Huang[1], John M Bekkers[1]\***

[1]Eccles Institute of Neuroscience, John Curtin School of Medical Research, The Australian National University, Canberra, Australia; [2]Neurocure Center for Excellence, Charité Universitätsmedizin Berlin and Humboldt Universität, Berlin, Germany

**Abstract** Feedforward inhibitory circuits are key contributors to the complex interplay between excitation and inhibition in the brain. Little is known about the function of feedforward inhibition in the primary olfactory (piriform) cortex. Using in vivo two-photon-targeted patch clamping and calcium imaging in mice, we find that odors evoke strong excitation in two classes of interneurons – neurogliaform (NG) cells and horizontal (HZ) cells – that provide feedforward inhibition in layer 1 of the piriform cortex. NG cells fire much earlier than HZ cells following odor onset, a difference that can be attributed to the faster odor-driven excitatory synaptic drive that NG cells receive from the olfactory bulb. As a result, NG cells strongly but transiently inhibit odor-evoked excitation in layer 2 principal cells, whereas HZ cells provide more diffuse and prolonged feedforward inhibition. Our findings reveal unexpected complexity in the operation of inhibition in the piriform cortex.

## Editor's evaluation

Feedforward inhibition (FFI) typically exerts a powerful effect shaping neural activity. In this paper, Suzuki et al., use a combination of in vivo and in vitro experiments to characterize, for the first time, responses in the two main classes of FFIs in the mouse olfactory cortex, neurogliaform cells (NG) and horizontal cells (HZ). They find that these two cell types have different responses and different connectivity, which partially explains their different responses. This paper also helps resolve a previously perplexing result from a recent publication proposing that FFI in the mouse olfactory cortex plays a negligible role in shaping cortical odor responses, presumably because those authors were only recording from HZ, but not NG, cells.

**\*For correspondence:**
John.Bekkers@anu.edu.au

**Present address:** †Institut fur Biologie, Berlin, Germany

**Competing interest:** The authors declare that no competing interests exist.

## Introduction

Synaptic inhibition in the cortex is an intricate process with many interacting parts. Cortical interneurons are classified into dozens of subtypes with distinctive morphological, transcriptomic, and electrophysiological properties (*Kawaguchi and Kubota, 1997*; *Tremblay et al., 2016*; *Zeng, 2017*; *Gouwens et al., 2020*). These diverse interneurons are embedded in at least three types of cortical inhibitory circuits – feedforward inhibition, feedback inhibition, and disinhibition (*Kullmann and Lamsa, 2007*; *Isaacson and Scanziani, 2011*; *Denève and Machens, 2016*; *Tremblay et al., 2016*; *Lourenço et al., 2020*). Within these three categories there exist numerous variants that implement a variety of neural computations. For example, feedforward and feedback inhibition exert different effects depending on whether the effector interneurons innervate the perisomatic or distal dendritic regions of their targets (*Bruno and Sakmann, 2006*; *Isaacson and Scanziani, 2011*; *Pardi et al., 2020*), and disinhibition can depend on the subtypes of interneurons involved and the origin of their inputs (*Pi et al., 2013*; *Fu et al., 2014*; *Tremblay et al., 2016*). In this paper we focus on feedforward

inhibition in the primary olfactory (piriform) cortex, and report that a surprising complexity of inhibitory processing is also a feature of this type of inhibition in the paleocortex.

Feedforward inhibition is conventionally understood as an input-tracking mechanism that does not depend on local circuit activity (*Tremblay et al., 2016*). In the primary neocortex, feedforward inhibition is often mediated by parvalbumin-expressing fast-spiking basket cells localized in the deeper layers that receive direct afferent input from first-order thalamic nuclei (*Beierlein et al., 2003*; *Gabernet et al., 2005*; *Cruikshank et al., 2007*). In this role feedforward inhibition improves sensory discrimination by favoring coincidence detection, but in addition it can provide gain modulation through input normalization (*Bruno and Sakmann, 2006*; *Carandini and Heeger, 2011*). More recently it has been reported that feedforward inhibition is also prominent in layer 1 of the neocortex, where it can engage afferent input from other cortical regions and from higher-order thalamic nuclei (*Pardi et al., 2020*; *Anastasiades et al., 2021*). However, the properties and functions of layer 1 neocortical interneurons have received relatively little attention and remain enigmatic (*Schuman et al., 2019*; *Fan et al., 2020*; *Lourenço et al., 2020*).

The piriform cortex (PCx) is unusual in that it receives all of its primary afferent input from the olfactory bulb into the superficial part of layer 1 (layer 1a; *Neville and Haberly, 2004*; *Bekkers and Suzuki, 2013*). Layer 1a mostly contains the apical dendrites of glutamatergic principal cells located in layers 2 and 3, as well as the somas, axons, and dendrites of two classes of GABAergic interneurons, namely, neurogliaform (NG) cells and horizontal (HZ) cells (*Suzuki and Bekkers, 2010a*). NG and HZ cells in layer 1a are thus ideally situated to provide feedforward inhibition onto the distal dendrites of layer 2/3 principal cells. Classic work (*Biedenbach and Stevens, 1969*; *Haberly, 1973*) as well as more recent studies (e.g. *Luna and Schoppa, 2008*; *Stokes and Isaacson, 2010*; *Suzuki and Bekkers, 2012*; *Sheridan et al., 2014*; *Stokes et al., 2014*; *Large et al., 2016a*; *Large et al., 2016b*) have used electrical stimulation to explore some of the basic properties of feedforward inhibition in the PCx. However, less is known about how particular types of interneurons respond to odor stimulation. The existence and importance of odor-evoked inhibition in the PCx has been established (*Poo and Isaacson, 2009*; *Zhan and Luo, 2010*; *Franks et al., 2011*; *Poo and Isaacson, 2011*; *Sturgill and Isaacson, 2015*; *Bolding and Franks, 2017*; *Tantirigama et al., 2017*; *Bolding and Franks, 2018*) but the identities and properties of neurons responsible for the different kinds of inhibition are, for the most part, uncertain.

Here, we approach this question by using two-photon-targeted whole-cell patch clamping and functional Ca$^{2+}$ imaging to record from visually identified NG and HZ cells in layer 1a of the PCx in vivo. We find that the odor-evoked feedforward inhibition provided by these two types of interneurons is strikingly different: NG cells express a powerful and transient inhibition that begins rapidly after odor onset, whereas HZ cells provide a more diffuse and delayed form of feedforward inhibition. Thus, two distinctive feedforward inhibitory circuits exist in the superficial layer of the PCx, where they are well placed to generate dynamically complex patterns of inhibition in the distal dendrites of principal cells.

## Results

### Interneurons that provide feedforward inhibition in the PCx can be targeted 'in vivo'

In this study we took advantage of the simple architecture of the PCx to examine in isolation only those neural circuits that generate feedforward synaptic inhibition. Afferent input from the olfactory bulb via the lateral olfactory tract (LOT) to the PCx is strictly confined to layer 1a (L1a) of the PCx (*Figure 1A*; *Neville and Haberly, 2004*). Hence, only those interneurons with dendrites concentrated in L1a, where they are able to intercept axons from the LOT, are able to generate feedforward inhibition (red cells, *Figure 1A*). Two such interneuron types have been identified in the PCx, HZ cells and L1a NG cells (*Suzuki and Bekkers, 2010a*). Conversely, feedback inhibition is largely mediated by interneurons with dendrites concentrated in deeper associational layers, two prominent types being fast-spiking (FS) and regular-spiking (RS) interneurons (blue cells, *Figure 1A*). In this paper we focus on the feedforward inhibitory neurons, HZ cells and L1a NG cells (here called 'NG cells' for short), located in the anterior PCx.

Pair recordings in slices show that all types of interneurons in the anterior PCx synaptically inhibit principal cells (*Figure 1A*; *Suzuki and Bekkers, 2010a*; *Suzuki and Bekkers, 2012*). Inhibition is

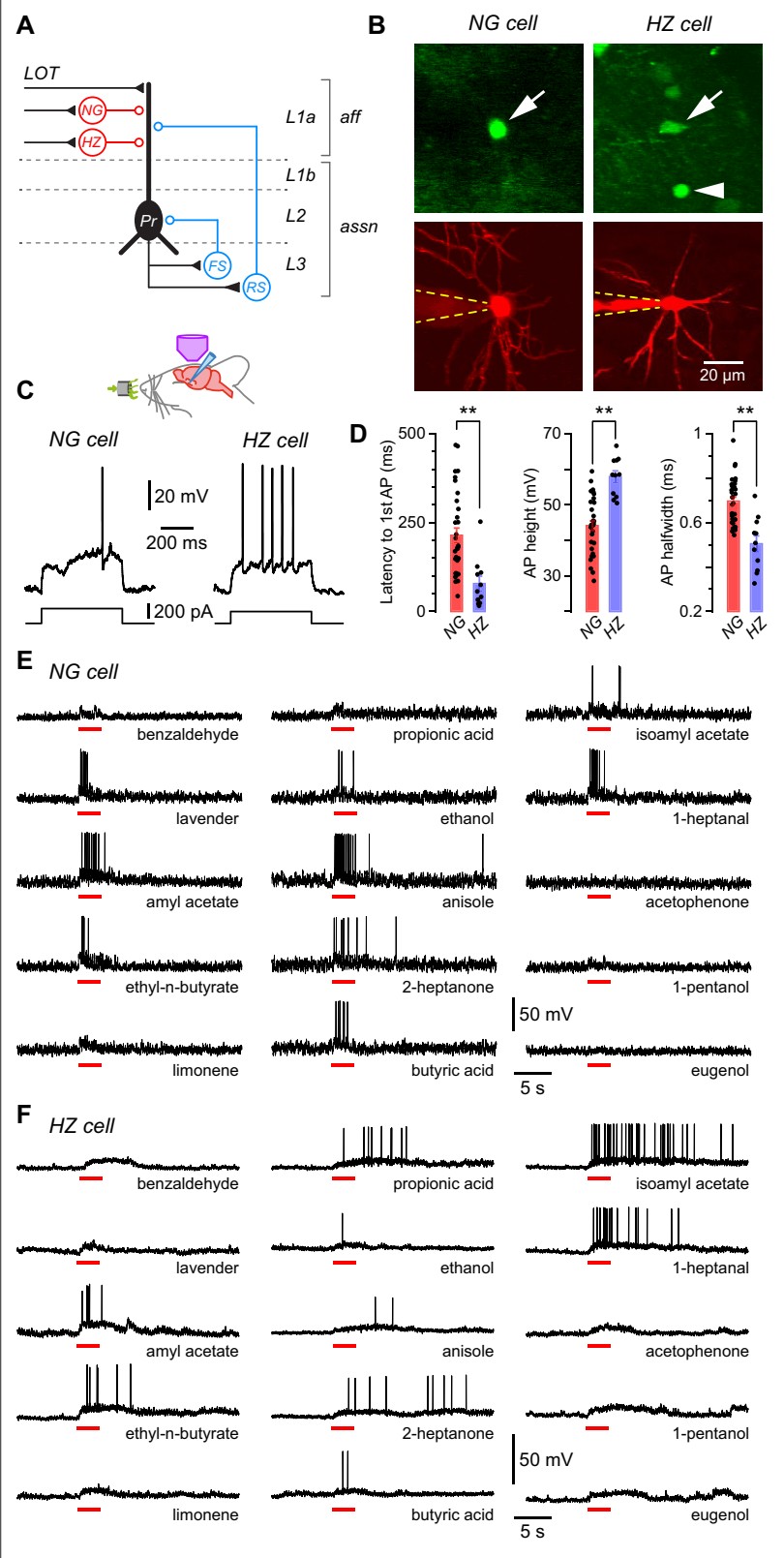

**Figure 1.** Layer 1a neurogliaform (NG) and horizontal (HZ) cells are reliably distinguished using two-photon-targeted patch clamping in vivo. NG and HZ cells respond differently to odors. (**A**) Schematic showing two forms of synaptic inhibition received by principal cells (Pr) in the PCx: feedforward (red, mediated by NG and HZ cells in layer 1a [L1a]) and feedback (blue, mediated by fast-spiking [FS] and regular-spiking [RS] cells in L3). Feedback

*Figure 1 continued on next page*

*Figure 1 continued*

inhibition from other cell types (e.g. bitufted cells, deep NG cells) is not shown. LOT, lateral olfactory tract; *aff*, afferent layer; *assn*, associational layers. (**B**) (Top) Two-photon images from a GAD67-GFP (Δneo) mouse showing (arrowed, left) the very bright GFP fluorescence in an NG cell and the much weaker GFP in an HZ cell (arrowed, right; arrowhead indicates a nearby NG cell). (**B**) (Bottom) z-Projection of the same two cells imaged in the red channel; internal solution contained Alexa Fluor-594. Dashed yellow lines show approximate position of patch electrode. Scale bar (bottom right) applies to all panels. (**C**) Responses of NG cell (left) and HZ cell (right) in vivo to a depolarizing current step near rheobase. (**D**) Comparison of selected properties of action potentials recorded in NG and HZ cells in response to current steps in vivo. Points show data from individual neurons, bars show mean ± standard error of the mean (SEM). **, $p < 0.01$, $n = 33$ cells for NG, $n = 11$ cells for HZ, Welch's two-sample unpaired t-test. (**E**) Response of an NG cell to a palette of 15 structurally diverse odors (name under each trace). Application period is shown by the red bar. Similar results were obtained from a total of 28 NG cells. (**F**) Recordings made from an HZ cell. Similar results were obtained from 10 HZ cells. Further details on analyzing these odor responses are in *Figure 1—figure supplement 1*.

The online version of this article includes the following source data and figure supplement(s) for figure 1:

**Source data 1.** Excel spreadsheet listing intrinsic electrical data shown in *Figure 1D*.

**Figure supplement 1.** Measurement of odor responsiveness.

commonly observed at the soma of principal cells in response to odors in vivo (e.g. *Poo and Isaacson, 2009*; *Zhan and Luo, 2010*; *Poo and Isaacson, 2011*; *Sturgill and Isaacson, 2015*; *Bolding and Franks, 2017*; *Tantirigama et al., 2017*; *Bolding and Franks, 2018*), although it is not immediately clear whether this inhibition is dominated by feedforward or feedback inhibitory circuits. We decided to explore feedforward inhibition in this study because the relevant interneurons (NG and HZ cells) are close to the cortical surface and more readily accessible to two-photon-targeted patch clamping in vivo.

In GAD67-GFP (Δneo) mice, NG and HZ cells can readily be distinguished under the two-photon microscope: HZ cells are only found close to the LOT whereas NG cells are distributed throughout L1a; HZ cell somata have an elongated shape whereas NG somata are spherical; and HZ cells express much lower levels of GFP than NG cells (*Figure 1B*, top; *Suzuki and Bekkers, 2010a*; *Suzuki and Bekkers, 2010b*). After whole-cell recording, their identities could be confirmed by their dendritic morphology (NG: short, thin, highly branched dendrites; HZ: longer, less branched, often spiny; *Figure 1B*, bottom) and distinctive intrinsic electrical properties (*Figure 1C and D*), similar to in vitro (*Suzuki and Bekkers, 2010a*). Thus, we were confident that we could record from identified NG and HZ cells in vivo.

## NG and HZ cells respond strongly to odors

We applied up to 15 odorants from a variety of chemical functional groups to urethane-anesthetized mice and measured the voltage responses of identified NG and HZ cells in the anterior PCx (*Figure 1E and F*). Three features were apparent in these responses: first, both cell types responded to many different odors (i.e. they were broadly tuned); second, regular oscillations in the membrane potential ($V_m$) often became larger in the presence of odor; and third, HZ cells appeared to respond more slowly to odors than did NG cells. Each of these features is examined in the following sections.

## NG and HZ cells are broadly excited by odors

Our palette of odorants was drawn from a variety of chemical functional groups intended to span a large part of 'odor space' (although this term is difficult to define; *Pashkovski et al., 2020*; *Ravia et al., 2020*). We measured odor responsiveness by median-filtering each trace to remove action potentials (APs) then testing whether the z-scored $V_m$ amplitude exceeded a positive threshold during the period of odor application (Experimental procedures; *Figure 1—figure supplement 1*). These data were analyzed in two different ways: (i) by calculating the fraction of odors that each cell responded to, then averaging across cells ('cell-averaged index'); and (ii) by calculating the fraction of cells each odor activated, then averaging across odors ('odor-averaged index'). NG and HZ cells both had large cell-averaged indices that were not significantly different from each other (NG: 0.78 ± 0.04, $n = 28$ cells from 19 mice; HZ: 0.66 ± 0.12, $n = 9$ cells from 9 mice; $p = 0.34$, Welch's two-sample unpaired t-test). The odor-averaged index was also large for both cell types but was significantly smaller for HZ cells

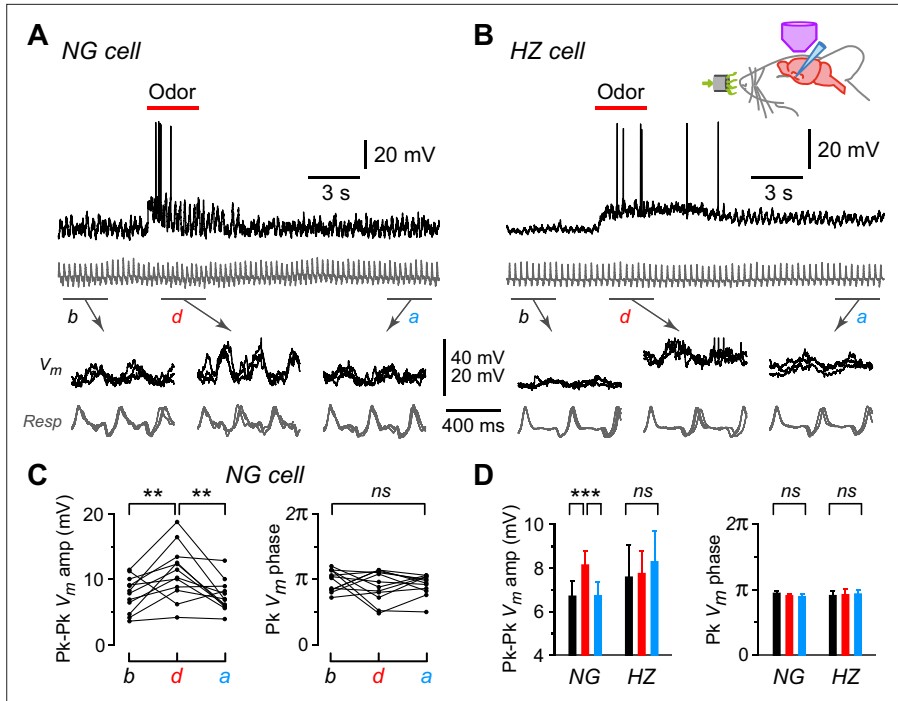

**Figure 2.** Oscillations in membrane potential ($V_m$) are synchronized to respiration in neurogliaform (NG) and horizontal (HZ) cells, and the oscillation amplitude is enhanced in NG cells during odors. (**A**) (Top) Response of an NG cell to ethyl-*n*-butyrate. Gray trace below shows the respiration. (**A**) (Bottom) Indicated time windows are shown expanded, with the traces in three consecutive sub-windows shown superimposed: *b*, before odor application; *d*, during; *a*, after. (**B**) Same for the response of an HZ cell to ethyl-*n*-butyrate. (**C**) Summary of measurements from the NG cell in panel (**A**) when applying 12 different odors. Plots show peak-to-peak amplitude of mean $V_m$ (left) and location of peak mean $V_m$ expressed as phase of respiration cycle (right), each averaged over ~10 respiration cycles occurring in a 3-s-long window before, during, or after odor application. Lines connect measurements made in the same sweep for each of the 12 odors. (**D**) Summary of mean peak-to-peak $V_m$ amplitudes (left) and phase (right) measured as in panel (**C**), averaged over *n* = 22 NG cells and *n* = 9 HZ cells. Black, red, and blue bars show mean ± standard error of the mean (SEM) before, during, and after odor application, respectively. \*\*\*, p < 0.001; \*\*, p < 0.01; *ns*, not significant; one-way ANOVA with Tukey. Similar results were obtained using an alternative approach, cross-covariance analysis (*Figure 2—figure supplement 1*). In contrast, respiration-synchronized oscillations in $V_m$ were not observed in an NG cell in the primary somatosensory cortex (*Figure 2—figure supplement 2*).

The online version of this article includes the following source data and figure supplement(s) for figure 2:

**Source data 1.** Excel spreadsheet listing peak-to-peak amplitude, cross-covariance, and phase data shown in *Figure 2D* and *Figure 2—figure supplement 1*, panel D.

**Figure supplement 1.** Cross-covariance analysis confirms the presence of respiration-locked oscillations in $V_m$ in neurogliaform (NG) and horizontal (HZ) cells, with the amplitude of these oscillations increasing in NG cells during odors.

**Figure supplement 2.** Subthreshold membrane potential ($V_m$) in neurogliaform (NG) cells in the primary somatosensory cortex (S1) does not oscillate in phase with respiration and does not respond to odors, in contrast to NG cells in the piriform cortex (PCx).

---

(NG: 0.75 ± 0.03, *n* = 15 odors; HZ: 0.60 ± 0.06, *n* = 15; p = 0.025, Welch's two-sample unpaired t-test). Similar results were found by using an alternative pair of measures, the lifetime and population sparseness, which are analogous to the cell- and odor-averaged indices, respectively (*Willmore and Tolhurst, 2001*; *Poo and Isaacson, 2009*).

## Respiration-locked oscillations in Vm are prominent in NG and HZ cells

Subthreshold oscillations in $V_m$ were frequently seen in both NG and HZ cells (e.g. *Figure 2A and B*, top, show the same cells as in *Figure 1E and F* responding to ethyl-*n*-butyrate; gray traces

are respiration, with upward transients indicating onset of exhalation). Expanding the traces in windows before (*b*), during (*d*), and after (*a*) odor application revealed that oscillations in $V_m$ were synchronized to respiration and, at least in the NG cell, the amplitude of $V_m$ oscillations appeared to increase during the odor (*Figure 2A and B*, bottom). These observations were quantified by excising the segments of $V_m$ that lay between successive positive peaks of the respiration trace, linearly warping them to have the same time axis, then averaging together all such segments within each of the three windows (*b*, *d*, *a*) for each odor. The peak-to-peak amplitude of the average $V_m$ and the time of the positive peak of average $V_m$, expressed as a phase of the respiration cycle, were plotted for each odor application (*Figure 2C*; each triplet of connected points is from each odor; only the data from the NG cell in *Figure 2A* are shown). For this particular NG cell, the peak-to-peak $V_m$ amplitude increased significantly during the odor (*b*: 7.7 ± 0.8 mV; *d*: 11.1 ± 1.2 mV; $n = 12$, p = 0.002, one-way ANOVA with Tukey's contrasts), whereas the phase of the peak was unchanged (*b*: 0.49 ± 0.02; *d*: 0.45 ± 0.03, both expressed as a fraction of the respiration cycle; p = 0.47).

A similar analysis was done for all NG and HZ cells in our dataset and the summary is shown in *Figure 2D*. The mean peak-to-peak $V_m$ amplitude increased significantly during odors in NG cells but not in HZ cells (*Figure 2D*, left; $n = 232$ trials in 22 NG cells, p < 0.0002; $n = 98$ trials in 9 HZ cells, p = 0.78; one-way ANOVA with Tukey). In contrast, the phase of the peak $V_m$ was unchanged by odors in both NG and HZ cells (*Figure 2D*, right). The same result was obtained if we used a different measure of phase-locking between $V_m$ and respiration, the cross-covariance (*Figure 2—figure supplement 1*). Thus, while NG and HZ cells both show strong respiration-coupled oscillations in their subthreshold $V_m$, odors affected these cells differently: in NG cells the oscillation amplitude increased, whereas in HZ cells $V_m$ tended to depolarize without a change in the oscillation.

We wondered if the oscillations in $V_m$ we found in NG (and HZ) cells in the PCx were also seen in NG cells in other cortical areas and, if so, whether they were synchronized to respiration. We made targeted whole-cell recordings from NG cells in the upper layers of primary somatosensory cortex in vivo and found that, although $V_m$ oscillations were strongly present, their temporal structure was different from those in PCx and they were not synchronized to respiration (*Figure 2—figure supplement 2*; see also *Suzuki et al., 2014*).

## NG and HZ cells tend to fire early and late, respectively, following odor onset

Next, we turned to the apparent difference in the kinetics of the odor response, with NG cells appearing to be excited more quickly following odor application (*Figure 1E and F*). We confirmed this impression by constructing AP raster plots and peristimulus time histograms (*Figure 3A*): spiking in NG cells reached a peak at 0.43 ± 0.14 s after odor onset ($n = 7$ cells), whereas in HZ cells the peak occurred at 1.92 ± 0.24 s ($n = 5$ cells; significantly different, p = 0.0013, Welch's two-sample unpaired t-test). Control experiments confirmed that none of these responses were limited by the rate of delivery of odorants by our olfactometer (*Figure 3—figure supplement 1*).

To test whether this difference in dynamics could be observed using a less invasive approach, we turned to two-photon $Ca^{2+}$ imaging with the red-shifted indicator Cal-590. NG and HZ cells could be distinguished as before from their soma location, soma shape and GFP fluorescence (*Figure 3B*, top), and odor-evoked spiking could be resolved from changes in $\Delta F/F_0$ (*Figure 3B*, bottom). Consistent with whole-cell patch clamping (*Figure 3A*), $Ca^{2+}$ imaging showed that NG cells fired quickly after odor onset and HZ cells fired with a delay (*Figure 3B*; traces are averages of $n = 6$ NG cells and $n = 5$ HZ cells; individual traces were too noisy to reliably determine the average time to peak).

We also took advantage of the $Ca^{2+}$ imaging approach to examine the effect of anesthetics on this neural circuit. (Because the surgery to expose the PCx is so invasive, our experiments could not use awake animals.) All of the above experiments used urethane (0.7 g/kg). We repeated the imaging experiment in *Figure 3B* using fentanyl plus medetomidine, which has been used to induce a more awake-like state of anesthesia (*Constantinople and Bruno, 2011*; *Altwegg-Boussac et al., 2014*). The result was the same (*Figure 3—figure supplement 2*): in response to odors, NG cells fired early, HZ cells late. These findings, together with a report that urethane at the relatively low concentration we used here has little effect on ligand-gated synaptic receptors (*Hara and Harris, 2002*), suggest that the different odor-response dynamics of NG and HZ cells are not related to anesthesia.

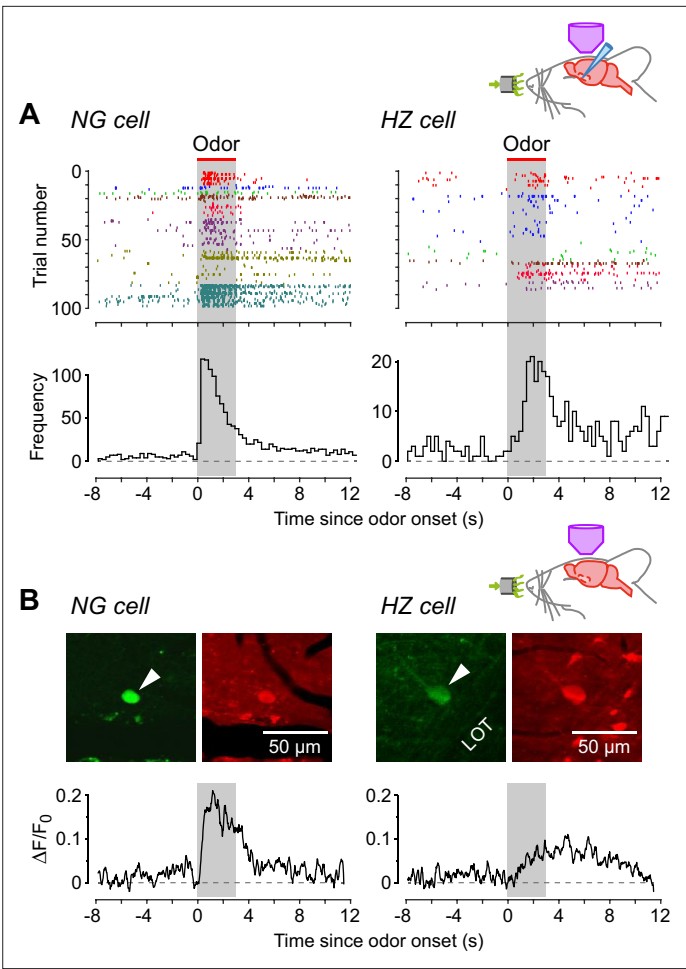

**Figure 3.** Neurogliaform (NG) cells tend to fire earlier than horizontal (HZ) cells in response to odors. (**A**) (Top) Spike raster plots for NG cells (left) and HZ cells (right) in our dataset that fired at least one AP before, during, or after odor application ($n$ = 98 trials in 8 NG cells, $n$ = 86 trials in 6 HZ cells). Different colors indicate different cells. Gray bar indicates period of odor application. (**A**) (Bottom) Peristimulus time histograms (PSTHs) for the raster plots, showing the greater delay to odor-evoked firing in HZ cells. These responses were not limited by the rate of delivery of odorants by our olfactometer (*Figure 3—figure supplement 1*). (**B**) (Top) Two-photon images of GFP (green) and Cal-590 (red) fluorescence for a layer 1a NG cell (left, arrowhead) and an HZ cell (right, arrowhead). (**B**) (Bottom) Averaged $\Delta F/F_0$ plots for the somatic responses of $n$ = 6 NG cells (left) and $n$ = 5 HZ cells (right) in response to odor. Similar results were observed using alternative anesthesia that produces a more awake-like state (*Figure 3—figure supplement 2*).

The online version of this article includes the following figure supplement(s) for figure 3:

**Figure supplement 1.** Response of horizontal (HZ) cells, but not neurogliaform (NG) cells, is much slower than the speed of odor arrival.

**Figure supplement 2.** In vivo two-photon calcium imaging experiments confirm that neurogliaform (NG) and horizontal (HZ) cells respond similarly to odors in animals anesthetized with either urethane or fentanyl and medetomidine.

## Different kinetics of odor-evoked EPSPs in NG and HZ cells

Having found that NG and HZ cells differ in their AP responses to odors, we next asked whether similar differences could be observed in the underlying EPSPs. Odor-evoked EPSPs were median-filtered to remove APs and notch-filtered at 2–4 Hz to remove respiration-coupled oscillations, then all EPSPs measured in the same cell for different odors were averaged together (responses to 4–15 odors per cell; *Figure 4—figure supplement 1*). Because these averages were still relatively noisy, a smooth curve (Materials and methods) was fitted to each averaged EPSP and the key parameters were

measured from this curve (*Figure 4—figure supplement 1*). The analysis showed that odor-evoked EPSPs reached their peak about twofold faster in NG cells than in HZ cells (*Figure 4A*; NG: 0.53 ± 0.05 s, *n* = 16 cells; HZ: 1.26 ± 0.26 s, *n* = 6 cells; p = 0.038, Welch's two-sample unpaired t-test). Thus, the slower-rising EPSPs in HZ cells (*Figure 4A*) can explain the longer delay to firing of these cells (*Figure 3*).

To confirm this result, we used a different method that focused on the rising phase of individual, unaveraged EPSPs. Traces were again median-filtered to remove APs, then the z-scored mean $V_m$ for each respiration cycle was plotted versus time and a straight line was fitted over the period of odor application (*Figure 4B*; typical NG cell on left, HZ cell on right). The slope of this line tended to be negative for NG cells and positive for HZ cells (red and blue line, respectively; *Figure 4B*, bottom). These data were quantified for all odors and cells by plotting the peak z-scored $V_m$ amplitude during the odor period versus the fitted slope (*Figure 4C*, top; NG cells in red, *n* = 310 trials; HZ cells in blue, *n* = 188 trials). Trials that did not give an odor response (i.e. with peak z-scored $V_m$ values ≤ 2.5) are grayed out (*Figure 4C*, top). Histograms of all the remaining trials confirm that the EPSP slopes for NG and HZ cells were significantly different (*Figure 4C*, bottom; NG slope, –0.27 ± 0.02, *n* = 232 trials above the z-score threshold; HZ slope, 0.21 ± 0.04, *n* = 84 trials above threshold; p < 0.001, Kolmogorov-Smirnov [KS] test). Thus, during odors, EPSPs in NG cells tend to decline from an early peak while EPSPs in HZ cells tend to rise to a later peak. These behaviors are consistent with the odor-evoked AP firing observed in NG and HZ cells (*Figure 3*).

## Differences in excitatory synaptic input can explain the odor response differences between NG and HZ cells

What cellular mechanisms might explain the different odor response dynamics of NG and HZ cells? Given the slow rise times of odor-evoked EPSPs in these cells (hundreds of milliseconds, much slower than electrically evoked EPSPs measured in slices; *Suzuki and Bekkers, 2010a*; *Suzuki and Bekkers, 2012*), we looked for ways in which slowly rising compound EPSPs in vivo could be constructed from trains of unitary EPSPs. Perhaps the simplest explanation is that NG cells receive depressing EPSPs from the olfactory bulb via the LOT, whereas HZ cells receive facilitating EPSPs (*Figure 5A*). To test this idea, we first recorded from NG and HZ cells in slices and applied patterned electrical stimuli to the LOT while pharmacologically blocking GABA$_A$ receptors. For brief trains of stimuli (6 pulses at 40 Hz, modeled on in vivo patterns; *Suzuki and Bekkers, 2006*), the response was the exact opposite of what was required: NG cells facilitated and HZ cells depressed (*Figure 5B*, top). We next tried a more realistic model of odor-evoked afferent excitation, that is, a train of 40 Hz stimuli repeating at 3 Hz (the approximate respiration frequency in mice; *Poo and Isaacson, 2009*). This stimulus produced a modestly depressing envelope of EPSPs in both NG and HZ cells (*Figure 5B*, bottom). Thus, the short-term dynamics of the LOT afferents could explain the declining response during odors in NG cells but not the facilitating response in HZ cells.

To further explore this question, we returned to in vivo recordings but now used whole-cell voltage clamp to isolate odor-evoked EPSCs at a holding potential of –70 mV (*Figure 5C*). Similar to the result for EPSPs (previous section), the time between odor onset and EPSC peak was about twofold faster in NG cells than in HZ cells (NG: 0.72 ± 0.09 s, *n* = 30 cell-odor pairs; HZ: 1.24 ± 0.12 s, *n* = 21 cell-odor pairs; p = 0.0014, Welch's two-sample unpaired t-test; *Figure 5C* inset, bottom). Thus, given that the membrane time constant is fast (~4 ms) and not significantly different between NG and HZ cells (*Suzuki and Bekkers, 2010a*), the slower EPSC rise time in HZ cells is consistent with the slower-rising EPSP (*Figure 4A*) and the delayed onset of odor-evoked spiking (*Figure 3*) in HZ cells.

In summary, despite the in vitro finding that HZ cells do not receive facilitating EPSPs from the LOT (*Figure 5B*), both current clamp (*Figure 4A*) and voltage clamp (*Figure 5C*) experiments in vivo show that HZ cells receive slower-rising synaptic inputs than NG cells. Thus, the dynamics of excitatory synaptic inputs at least partially determine the delayed odor responses of HZ cells, although the origin of these slower dynamics remains unclear (see Discussion).

## Synaptic inhibition also contributes to the odor responses of NG and HZ cells

Although we have so far focused on excitatory synaptic inputs, a likely contribution of inhibitory inputs cannot be excluded. NG and HZ cells may engage in mutual inhibition, thereby modifying the

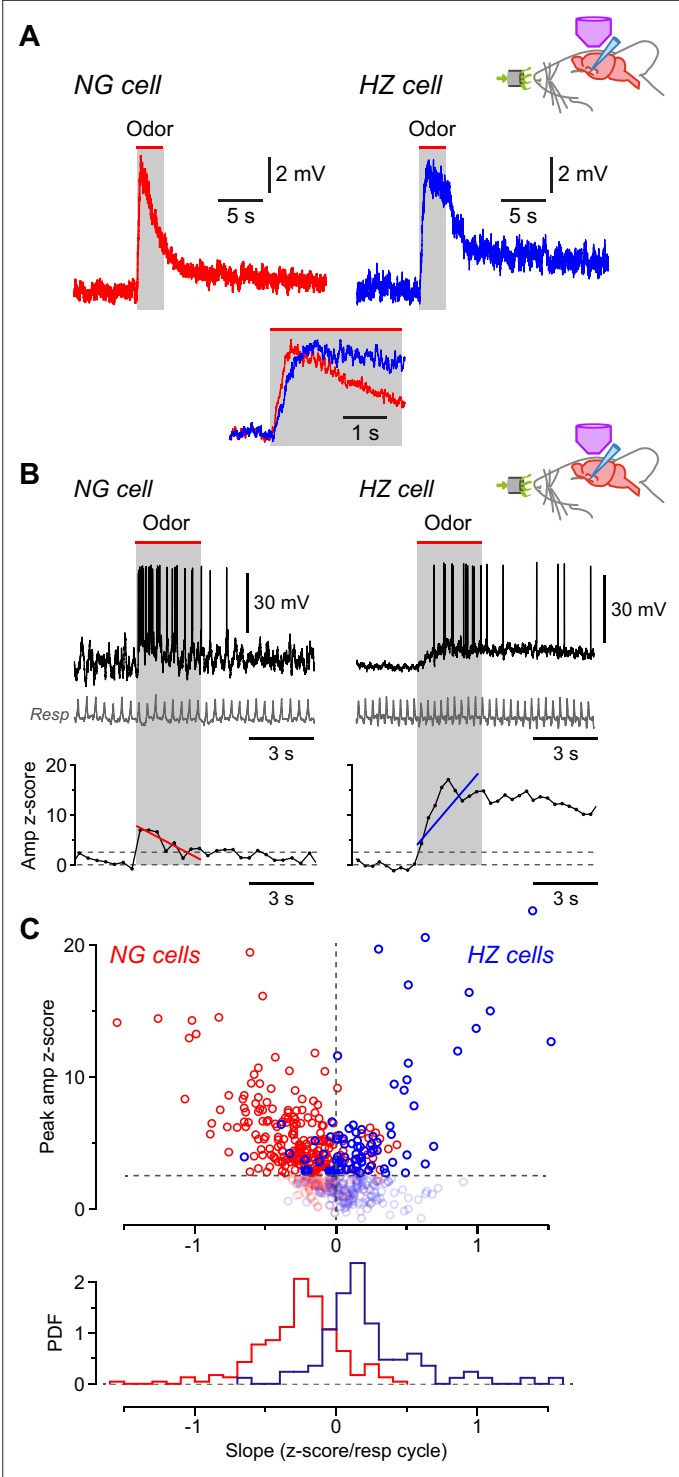

**Figure 4.** Odors cause $V_m$ to depolarize more slowly in horizontal (HZ) cells than in neurogliaform (NG) cells. (**A**) Averaged in vivo current clamp recordings of odor-evoked EPSPs measured in NG cells (left, $n$ = 16 cells) and HZ cells (right, $n$ = 6). Inset (bottom) shows the same traces normalized to their peaks, overlaid and expanded, showing that averaged EPSPs in HZ cells have a slower time to peak. To facilitate comparison, these EPSPs were filtered to remove respiration-linked oscillations (**Figure 4—figure supplement 1**). (**B**) (Top) Response of an NG cell to ethyl-*n*-butyrate (left) and an HZ cell to 1-heptanal (right). The corresponding respiration trace is shown below (gray). (**B**) (Bottom) z-Score-transformed mean $V_m$ amplitude (averaged over each respiration cycle) calculated for the same two neurons as above. Spikes were removed before measuring $V_m$. Upper horizontal

*Figure 4 continued on next page*

*Figure 4 continued*

dashed line indicates the detection threshold (z-score = 2.5) for an odor-evoked response. Superimposed red (left) or blue (right) line represents a linear fit to the data points over the 3-s-long odor application period (gray band), giving slopes of –0.70 and 1.36 z-score units/respiration cycle, respectively. (**C**) (Top) Plot of peak $V_m$ z-score during odor application versus slope (fitted as in panel **B**, bottom), where each data point represents the odor response of a single cell (red symbols, $n$ = 310 responses from 28 NG cells; blue symbols, $n$ = 188 responses from 9 HZ cells). Points below the threshold for an odor-evoked response (z-score = 2.5) have been grayed out. (**C**) (Bottom) Probability density function (PDF) of the points plotted in panel (**C**) (top), excluding the grayed-out values. NG cells (red plot) show a significant skew toward negative slopes, whereas HZ cells (blue) skew toward positive slopes (p < 0.001; Kolmogorov-Smirnov test).

The online version of this article includes the following source data and figure supplement(s) for figure 4:

**Source data 1.** Excel spreadsheet listing z-score-transformed peak $V_m$ during odor application and slope, shown in *Figure 4C*.

**Figure supplement 1.** Procedure for measuring the properties of odor-evoked EPSPs.

odor-evoked excitation both receive from the LOT. We addressed this possibility by making in vivo whole-cell voltage clamp recordings to look for odor-evoked IPSCs in isolation at a holding potential of +10 mV (*Poo and Isaacson, 2009*). Such IPSCs were, indeed, present in both HZ and NG cells (*Figure 6A*). Their properties were not significantly different between the two cell types (mean amplitude: 50.3 ± 4.3 pA versus 49.9 ± 11.0 pA; time to peak: 0.58 ± 0.12 s versus 0.35 ± 0.05 s; data for NG and HZ cells, respectively; all p > 0.1, $n$ = 14 and 5 cell-odor pairs, respectively). Current clamp experiments confirmed that odor-evoked IPSPs could also be observed in HZ cells when they were depolarized by somatic current injection (*Figure 6—figure supplement 1*).

What might be the origin of these inhibitory inputs? We addressed this question by making pair recordings in slices. These experiments revealed that NG→HZ cell connections were frequent and strong (*Figure 6B*; 14/43 = 32.6% connectivity; mean connection conductance 0.87 ± 0.41 nS, range 0.10–5.86 nS, $n$ = 14 pairs) whereas the reverse connection was never seen (0/13 = 0%; significantly different, p = 0.02, chi-square 2 × 2 contingency test). NG→NG cell connectivity (2/17 = 11.8%) was not significantly different from NG→HZ connectivity (p = 0.1, chi-square 2 × 2 contingency test; *Figure 6B*, right) although the $n$-values are modest. In contrast, connectivity between HZ cells was weaker (1/19 = 5.3%; *Figure 6B*, right). To sum up, these connectivity experiments suggest that odor-evoked synaptic inhibition of both NG and HZ cells most likely comes from NG cells after they are excited by the LOT.

If NG cells are responsible for much of the inhibition of layer 1a interneurons, do NG cells also receive privileged input from the LOT? We addressed this question by making dual whole-cell recordings from an NG and HZ cell in a slice while applying a train of minimal electrical stimuli (five at 20 Hz) to the LOT while pharmacologically blocking GABA$_A$ receptors (*Figure 6C*). In some of these experiments (4 out of 7 pairs), a clear plateau region was seen in a plot of EPSC amplitude versus stimulus number as the stimulus strength was progressively increased in small steps (*Figure 6C*, bottom left, stimulus numbers 20–45); this is suggestive of a unitary input (*Stokes and Isaacson, 2010*). Notably, the EPSC in the NG cell was always much smaller than that in the HZ cell for the first EPSC in the train (e.g. *Figure 6C*, bottom left and top right). This result did not depend on which cell was closer to the LOT or stimulating electrode (data not shown). A similar effect was seen in the three experiments in which a plateau was not observed (e.g. *Figure 6—figure supplement 2*).

Interestingly, because of the characteristic short-term facilitation and depression of LOT inputs to NG and HZ cells, respectively (*Suzuki and Bekkers, 2010a*), the imbalance in the strength of LOT inputs to these two cells changed during the train. This effect was quantified by calculating the ratio of the EPSC amplitude in the NG cell to that in the HZ cell for each of the five EPSCs in the train (*Figure 6C*, bottom right). For the first EPSC, the mean NG:HZ amplitude ratio was 0.27 ± 0.03 (mean ± standard error of the mean [SEM], $n$ = 7 cell pairs), but by the third EPSC this ratio had increased to 1.38 ± 0.32 (significantly different, p = 0.0002, $n$ = 7, one-way ANOVA with Tukey; *Figure 6C*, bottom right). These results show that NG cells do not receive privileged input from the LOT compared to HZ cells; rather, the reverse is the case for the first EPSC in a train. Later in the train, however, the inputs become more equal. Given the known dynamics of afferent input from the olfactory bulb (i.e.

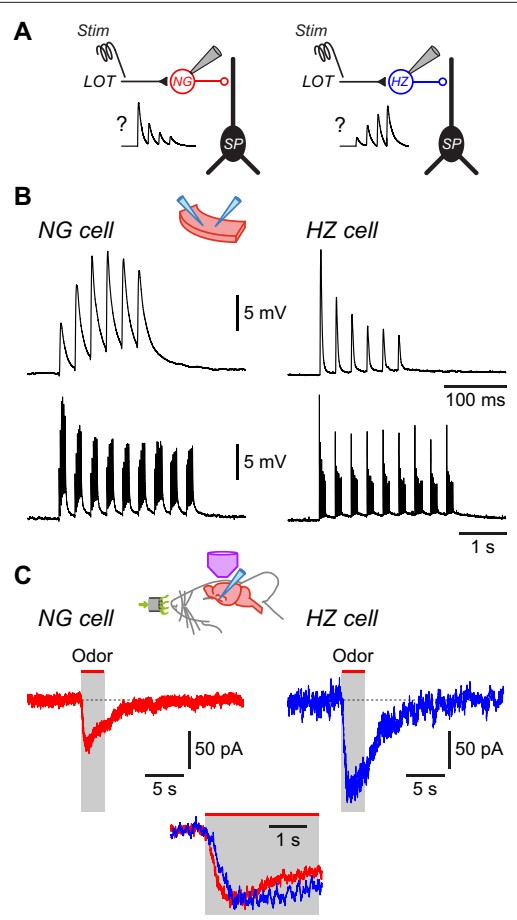

**Figure 5.** Differences in EPSC kinetics partly account for the different odor responsiveness of neurogliaform (NG) and horizontal (HZ) cells. (**A**) Schematic showing the circuit dynamics that are proposed to explain the results in **Figure 4**, viz. lateral olfactory tract (LOT) input onto NG and HZ cells is hypothesized to be depressing or facilitating, respectively, as illustrated in the schematic EPSP traces labelled '?'. (**B**) Slice experiments showing EPSPs elicited in NG cells and HZ cells in response to trains of extracellular stimuli applied to the LOT with 100 µM picrotoxin in the bath to block GABA$_A$ receptors. (**B**) (Top) Six stimuli at 40 Hz elicit facilitating EPSPs in an NG cell (left) but depressing EPSPs in an HZ cell (right), which is opposite to the hypothesized responses (panel **A**).(**B**) (Bottom) A longer train (9 bursts of 40 Hz trains, repeated at 3 Hz), thought to more closely replicate odor-evoked stimuli, elicits a depressing envelope of EPSPs in both an NG cell (left) and an HZ cell (right). Thus, the hypothesis in panel (**A**) is supported for NG cells but not for HZ cells. Stimulus artifacts are blanked. Similar results were obtained in $n = 6$ experiments of this kind for each cell type. (**C**) Averaged in vivo voltage clamp recordings of odor-evoked EPSCs measured in NG cells (left, $n = 17$ cell-odor pairs) and HZ cells (right, $n = 16$ cell-odor pairs). Holding potential –70 mV. Inset (bottom) shows the same traces overlaid and expanded, showing that averaged EPSCs in HZ cells have a slower time to peak.

modulated bursts of APs; *Cang and Isaacson, 2003*; *Margrie and Schaefer, 2003*), these results suggest that NG and HZ cells mostly operate in a regime where they receive similar afferent excitation.

## Feedforward inhibition alters the synaptic responsiveness of SP cells

Lastly, we explored the functional consequences of HZ and NG cell inhibition for one of their major targets, layer 2 superficial pyramidal (SP) cells. Because it is difficult to disambiguate these two types of feedforward inhibition in vivo, we conducted the experiments in slices.

We began by eliciting in vivo-like IPSPs in SP cells. Extracellular stimuli were applied to layer 1a in patterns obtained from the in vivo odor-evoked firing patterns of NG and HZ cells (*Figure 7A*, red traces; gray bars labelled 'odor period' represent the period during which the odor was applied in the in vivo experiments). Averaged postsynaptic IPSPs were recorded in SP cells while pharmaco-logically blocking ionotropic glutamate receptors (*Figure 7A*, black traces; average of $n = 44$ or 51 single traces for 13 or 3 different NG or HZ cell stimulus patterns, respectively, while recording from 6 different SP cells).

Two further manipulations were done to make the recordings more in vivo-like. First, NG stimulus recordings were made distant (>400 µm) from the LOT to avoid stimulating HZ cell axons, which are clustered around the LOT (*Suzuki and Bekkers, 2010a*). On the other hand, HZ stimulus record-ings were made near the LOT, where a mixture of HZ and NG cell axons were likely excited. Second, we warped the stimulus patterns so the respiration trace recorded for each stimulus pattern matched a reference respiration trace (shown in gray, *Figure 7A*, bottom; see Materials and methods; a similar approach was used in *Figure 2*). This warping of the time base was done to preserve any respiration-synchronized structure in the stim-ulus patterns (as in *Figure 2*) when averaging across different patterns. Such synchronization is apparent in the respiration-locked oscillations in the averaged IPSPs (*Figure 7A*, black traces). These results show that NG cells generate a large, rapid IPSP in SP cells, whereas HZ cells generate a smaller and more diffuse IPSP that persists beyond the end of odor application.

In a final series of experiments we examined the effect of these two types of synaptic inhibi-tion on spiking patterns in postsynaptic SP cells. The method described in the previous paragraph was employed, except that a dynamic clamp was

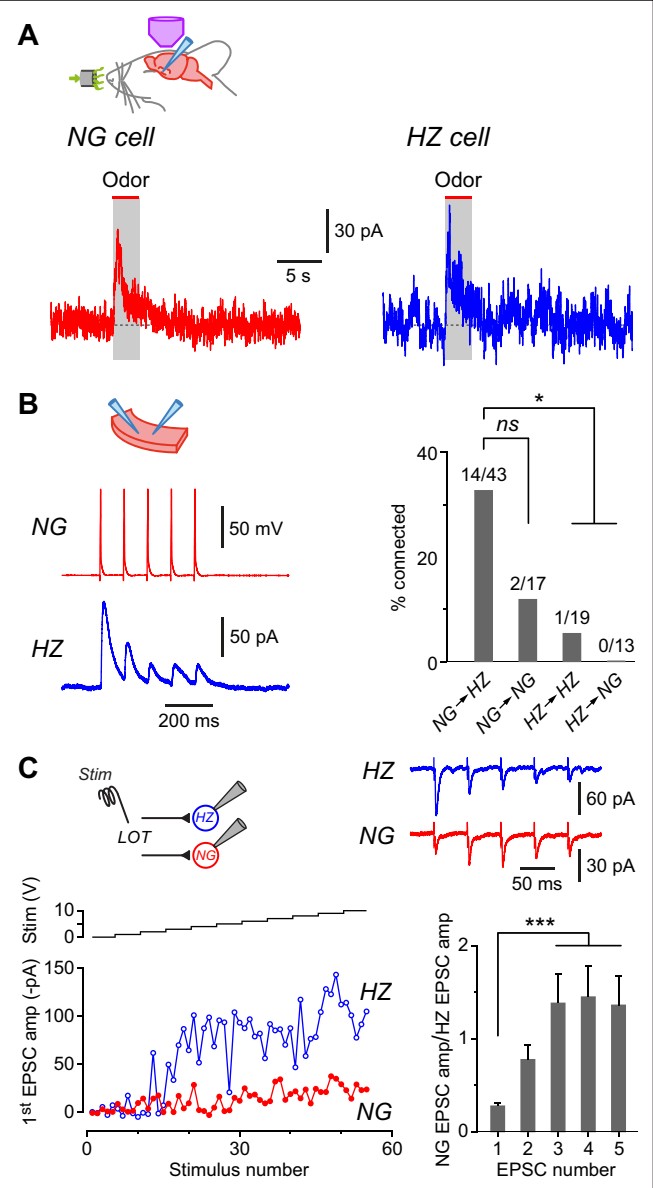

**Figure 6.** Neurogliaform (NG) and horizontal (HZ) cells receive odor-evoked feedforward inhibition, predominantly from other NG cells. (**A**) Averaged in vivo voltage clamp recordings of odor-evoked IPSCs measured in NG cells (left, *n* = 14 cell-odor pairs) and HZ cells (right, *n* = 5 cell-odor pairs). Holding potential +10 mV. In separate experiments it was found that synaptic inhibition could also be observed in HZ cells under current clamp (*Figure 6—figure supplement 1*). (**B**) (Left) Dual recording from a synaptically connected NG cell and HZ cell pair in a slice. Stimulus was a 20 Hz train of 2-ms-long depolarizing current steps applied to the NG cell. Postsynaptic response is an average of 15 episodes (holding potential, 0 mV). (**B**) (Right) Summary bar plot showing percentage of NG/HZ cell pairs tested in vitro that were synaptically connected. Numbers above bars indicate 'number of connections/total number of pairs tested'. *, p < 0.05; *ns*, not significant; chi-square 2 × 2 contingency test. (**C**) (Top left) Schematic showing the recording configuration for this panel, with a stimulating electrode in the lateral olfactory tract (LOT) and dual whole-cell recordings from an NG cell and an HZ cell in a slice. (**C**) (Bottom left) Example experiment showing peak EPSC amplitude for the first EPSC in a train of five recorded simultaneously in NG and HZ cells plotted versus stimulus number (red and blue traces, bottom) while increasing the stimulus strength in 1 V steps ('Stim', black staircase, top). (**C**) (Top right) Example data for the experiment shown in bottom left. Each trace is the average of responses to stimulus numbers 30–39. Stimulus artifacts are truncated. (**C**) (Bottom right) Summary bar plot showing the mean ± standard error of the mean (SEM) of NG:HZ EPSC amplitude ratios for all recorded pairs calculated for each EPSC in the train. ***, p < 0.001, *n* = 7 dual recordings in 7 slices from 4 mice.

*Figure 6 continued on next page*

*Figure 6 continued*

The online version of this article includes the following figure supplement(s) for figure 6:

**Figure supplement 1.** Steady depolarization of a horizontal (HZ) cell in vivo reveals odor-evoked hyperpolarization and suppression of firing.

**Figure supplement 2.** The lateral olfactory tract (LOT)-evoked EPSC recorded in a horizontal (HZ) cell is larger than that recorded simultaneously in a neurogliaform (NG) cell even when a 'plateau' is not apparent in the stimulus-response plot.

used to replay into the SP cell an odor-evoked excitatory postsynaptic conductance (EPSG) that had previously been recorded from an SP cell in vivo (*Figure 7—source code 1*). Again, the EPSG and all stimulus patterns were warped to match their respiration traces. When injecting the EPSG alone, an in vivo-like train of APs was evoked in the SP cell (*Figure 7B*, black traces). When patterned extracellular stimuli were applied at the same time, eliciting IPSPs, fewer APs were evoked in the SP cell by the EPSG (*Figure 7B*, blue traces; spikes that dropped out are indicated by red asterisks above the black traces). These effects were quantified by calculating normalized PSTHs for APs in the SP cell (*Figure 7B*, bottom; black, EPSG alone; blue, EPSG plus IPSPs; averages from $n$ = 7 SP cells with NG stimulus patterns, or $n$ = 4 SP cells with HZ stimulus patterns). The results confirm that NG cells strongly suppress early odor-evoked firing in SP cells ($p < 0.001$, KS test), whereas HZ cells tend to have a weaker, delayed effect (here, not significant; $p = 0.81$, KS test).

## Discussion

In this paper we used whole-cell patch clamping and two-photon $Ca^{2+}$ imaging in vivo to characterize the odor responses of two types of GABAergic interneurons that provide feedforward inhibition in the input layer (layer 1a) of the anterior PCx. We find that both NG cells and HZ cells are broadly excited by different odors, but the time to reach peak excitation is much slower in HZ cells than in NG cells. This difference can be explained by a twofold slower time to peak of the odor-evoked compound EPSC in HZ cells compared to NG cells. Synaptic inhibition of HZ cells by NG cells may also help to suppress early firing of HZ cells, and suggests how lateral inhibition and feedforward inhibition may interact in the same circuit. In addition, these two cell types differ in their oscillatory response to odors, with NG cells showing larger-amplitude respiration-coupled EPSPs during odor sampling. Lastly, we show that NG and HZ cells have distinctive effects on the excitability of downstream pyramidal neurons: NG cells generate powerful inhibition immediately after odor onset, whereas HZ cells exert more diffuse and prolonged inhibition. Our findings reveal different types of inhibitory responses at the first stage of cortical odor processing, and add to a growing understanding of the role of afferent-driven feedforward inhibition in cortical processing more broadly (*Khubieh et al., 2016*; *Fan et al., 2020*; *Anastasiades et al., 2021*).

### Use of anesthesia

Two-photon-targeted patch clamping and $Ca^{2+}$ imaging as implemented in this study required direct access to the surface of the PCx (*Margrie et al., 2003*; *Tantirigama et al., 2017*). Although similar experiments can be done during wakefulness and semi-paralysis of mice (*Pashkovski et al., 2020*), our animal ethics protocols required us to conduct all surgery and experiments under general anesthesia. We used urethane at the minimum concentration empirically determined to be effective in providing stable anesthesia with complete abolition of reflexes (0.7 g/kg *s.c.*). Urethane has been widely used in previous in vivo studies of the PCx (e.g. *Barnes et al., 2008*; *Poo and Isaacson, 2009*; *Poo and Isaacson, 2011*; *Wesson et al., 2011*; *Chapuis and Wilson, 2011*; *Sturgill and Isaacson, 2015*), and we have confirmed that urethane at this concentration has no effect on electrical activity in the PCx compared with mice anesthetized with fentanyl/medetomidine (which induces a more awake-like brain state; *Figure 3—figure supplement 2*; *Constantinople and Bruno, 2011*; *Tantirigama et al., 2017*). We avoided using ketamine/xylazine anesthesia which has been reported to alter PCx activity through its blockade of NMDA receptors (*Fontanini and Bower, 2005*; *Tantirigama et al., 2017*). Despite these considerations, however, it remains possible that our findings are affected by urethane and further study is warranted.

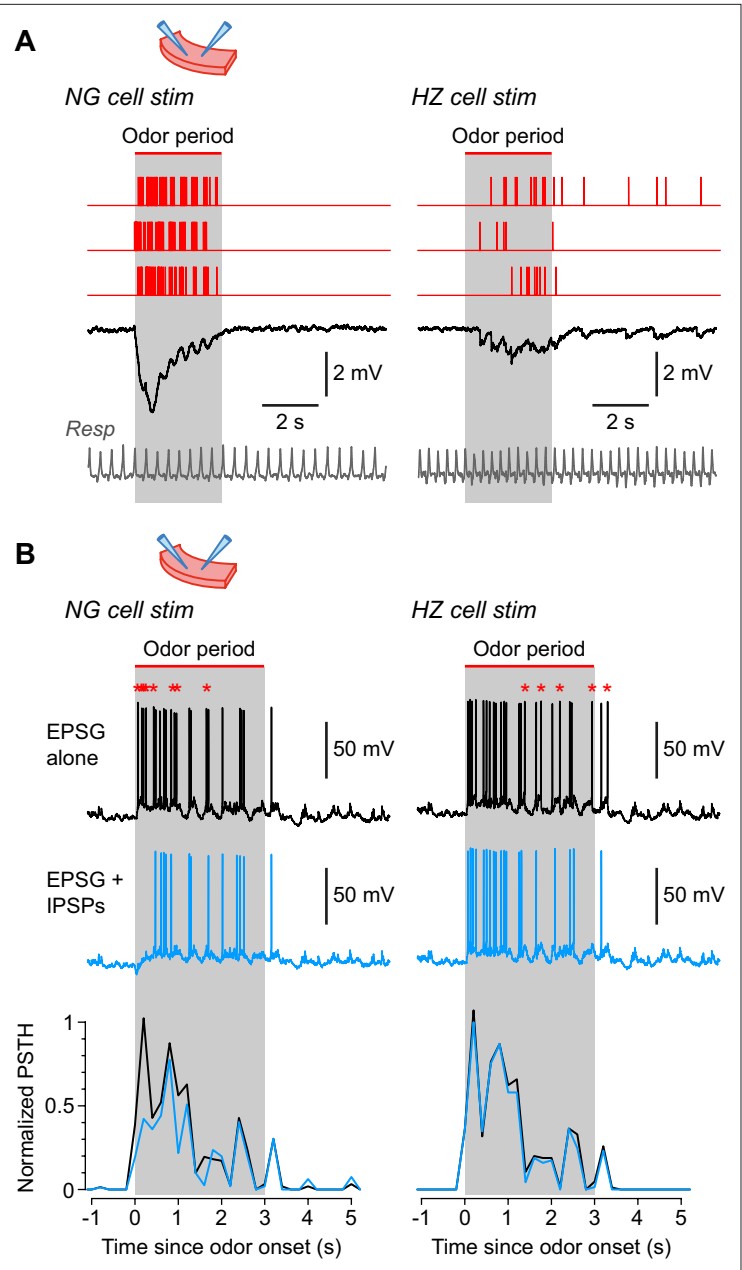

**Figure 7.** Neurogliaform (NG) and horizontal (HZ) cells produce different patterns of synaptic inhibition in postsynaptic superficial pyramidal (SP) cells. (**A**) Average IPSPs recorded in vitro in an SP cell (black traces; stimulus artifacts blanked) in response to extracellular stimulation in layer 1a of the slice, using spiking patterns that had previously been recorded in vivo in either NG cells (left) or HZ cells (right). Examples of these spiking patterns are shown in the red traces (top). IPSPs were averaged from *n* = 6 SP cells, to each of which was applied up to 16 different stimulus patterns (NG and HZ). To preserve possible respiration-synchronized effects (*Figure 2*) when averaging across different stimulus patterns, each raw IPSP trace was 'warped' to match a reference respiration trace (gray, bottom). (**B**) (Top) Example action potentials recorded in SP cells while using a dynamic clamp to inject an odor-evoked excitatory postsynaptic conductance (EPSG) that had previously been recorded in vivo (data in *Figure 7—source code 1*). Responses with the EPSG alone (upper traces, black) were interleaved with responses to the EPSG plus extracellular synaptic stimulation (lower traces, blue; stimulus artifacts blanked) driven by different odor-evoked spiking patterns for NG cells (left) or HZ cells (right). As in panel (**A**), stimulus patterns were warped to a reference respiration trace (*Figure 7—source code 1*). Asterisks above the upper traces label action potentials that are absent in the lower traces. All slice recordings in panels (**A**) and (**B**) were made in the presence of 20 µM CNQX and 25 µM D-AP5. (**B**) (Bottom) Mean normalized PSTH plots from *n* = 7 SP cells showing

*Figure 7 continued on next page*

*Figure 7 continued*

that synaptic inhibition patterned on NG cell activity significantly suppresses early firing in SP cells (left), whereas inhibition patterned on HZ cell firing has a weak, delayed effect (right). Gray bars labelled 'odor period' in panels (**A**) and (**B**) indicate the period during which odor was applied in the in vivo recordings from NG or HZ cells. These recordings were used to construct the in vivo-like stimulus patterns (sample data in ***Figure 7—source code 1***).

The online version of this article includes the following source code for figure 7:

**Source code 1.** Igor procedure file containing code used to acquire the data in Figure 7, as well as sample data and a ReadMe file explaining how to install and run the code.

## Comparison with previous findings

Most previous work has used electrical stimulation in slices to study feedforward inhibition in the PCx (***Luna and Schoppa, 2008***; ***Stokes and Isaacson, 2010***; ***Suzuki and Bekkers, 2010a***; ***Suzuki and Bekkers, 2012***; ***Sheridan et al., 2014***; ***Stokes et al., 2014***; ***Large et al., 2016a***; ***Large et al., 2016b***). Only two reports, to our knowledge, have explicitly examined this feedforward circuit using odor stimulation in vivo. ***Poo and Isaacson, 2009***, made blind whole-cell recordings from neurons in layer 1, while ***Bolding and Franks, 2018***, made unit recordings from optogenetically identified GABAergic neurons in layer 1. However, neither study distinguished interneuron subtypes.

We found broad tuning of odor-evoked EPSPs in NG and HZ cells, consistent with previous findings (***Poo and Isaacson, 2009***; odor-averaged index, 0.50 ± 0.04 *cf* 0.75 and 0.60 for NG and HZ cells, respectively, found here). Interestingly, broad odor tuning has also been reported for some (***Zhan and Luo, 2010***; ***Poo and Isaacson, 2011***; ***Bolding and Franks, 2017***) but not all (***Sturgill and Isaacson, 2015***) classes of feedback inhibitory neurons located in deeper layers of the PCx. It is likely that this diversity in stimulus tuning reflects the number and variety of functional inputs from upstream neurons. Indeed, it has been shown that layer 1 interneurons receive a higher convergence of afferent input from the olfactory bulb than do principal neurons (***Poo and Isaacson, 2009***; ***Miyamichi et al., 2011***). Future work could repeat these experiments while distinguishing NG and HZ cells. Differences might arise because HZ cells, unlike NG cells, are clustered around the LOT where they may encounter a higher density of afferents (***Suzuki and Bekkers, 2010a***).

We often observed oscillations in subthreshold $V_m$ that were phase-locked to respiration, consistent with previous reports that such oscillations are ubiquitous in the olfactory system (***Fontanini and Bower, 2006***; ***Kay et al., 2009***; ***Poo and Isaacson, 2009***; ***Wilson, 2010***; ***Oswald and Urban, 2012***; ***Kay, 2014***; ***Jiang et al., 2017***). Our novel finding was that respiratory oscillations in $V_m$ often increased in amplitude during odor sampling, but only in NG cells. These odor-dependent oscillations were functionally relevant because they generated respiratory-patterned IPSPs in target neurons (***Figure 7A***) and provided rhythmic inhibition of similarly patterned EPSPs (***Figure 7B***; see also ***Poo and Isaacson, 2009***). This difference between NG and HZ cells suggests that NG cells provide fast, phase-critical feedforward inhibition whereas HZ cells provide a slower, tonic form of inhibition. Interestingly, ***Bolding and Franks, 2018***, reported only the latter kind of feedforward inhibition in multi-unit recordings from awake mice.

## Difference in odor-evoked EPSP kinetics

Our main finding is that odor-evoked firing in HZ cells has a delayed onset, contrasting with the rapid onset in NG cells. We showed that this effect can be at least partly explained by the slower time-to-peak of the odor-evoked compound EPSC in HZ cells (***Figure 5C***). Surprisingly, however, these kinetic differences cannot be explained by the properties of short-term synaptic plasticity at LOT synapses, assayed in slices (***Figure 5B***). What might be alternative explanations?

The LOT is a heterogeneous fiber tract that contains the axons of two distinct types of projection neurons in the olfactory bulb, mitral and tufted cells. Mitral cells have been reported to respond about twofold more slowly to odor stimulation than tufted cells (***Fukunaga et al., 2012***; ***Igarashi et al., 2012***) because of differences in synaptic inhibition in the bulb (***Fukunaga et al., 2012***; ***Geramita and Urban, 2017***). An appealing possibility is that mitral and tufted cell axons preferentially target HZ and NG cells, respectively; that is, the delayed odor response of HZ cells may originate in the olfactory bulb rather than at synaptic terminals in the PCx. A difficulty with this idea is that the axons of tufted

cells in the dorsal olfactory bulb do not extend far from the LOT and only reach about halfway along the length of the anterior PCx, that is, to ~1 mm anterior to Bregma (*Paxinos and Franklin, 2001*; *Igarashi et al., 2012*). Our in vivo recordings were made close to the LOT but more caudally, around 0.6 mm anterior to Bregma. However, it remains possible that tufted cells in other parts of the olfactory bulb (not just in the dorsal bulb) have a more caudal projection. This hypothesis could be tested in future work.

A second possible explanation is that HZ cells may receive delayed bulbar input via an intermediary, such as the anterior olfactory nucleus (AON; *Haberly and Price, 1978*; *McGinley and Westbrook, 2011*; *Kay and Brunjes, 2014*). Although AON axons are sparser in layer 1a than in deeper layers, they are consistently seen there (*Haberly and Price, 1978*; *Hagiwara et al., 2012*; *Russo et al., 2020*). This hypothesis could be tested anatomically or by silencing the AON during odor application.

A third possibility is that lateral inhibition from NG cells counteracts early excitation in HZ cells, delaying depolarization. In support of this idea, NG cells profusely innervate HZ cells (*Figure 6B*) and an odor-evoked hyperpolarization with the right time course can be observed in HZ cells (*Figure 6— figure supplement 1*). In addition, NG cells receive facilitating excitatory input from the LOT, driving them at least as strongly as HZ cells during the bursts of spikes that typify odor-evoked output from the olfactory bulb (*Figure 6C*; *Cang and Isaacson, 2003*; *Margrie and Schaefer, 2003*). On the other hand, NG cells also receive odor-evoked synaptic inhibition, presumably from other NG cells (*Figure 6A*), yet do not exhibit delayed odor responses like HZ cells. It is plausible that inhibition from NG cells is larger in HZ cells but it was not apparent here because the number of experiments was relatively small.

It should be kept in mind that both NG and HZ cells may also receive feed*back* inhibition from interneurons located in deeper layers, particularly from layer 3 somatostatin-positive RS interneurons with axons that ramify in L1 (*Suzuki and Bekkers, 2010a*). HZ cells can extend their dendrites into L1b (*Suzuki and Bekkers, 2010a*) so might be especially susceptible to feedback inhibition of this kind. Thus, depending on patterns of activity in afferent and associational circuits, feedforward and feedback inhibition may interact.

## Functional significance of feedforward inhibition in the PCx

Feedforward inhibition is generally recognized as an input-tracking mechanism which, in the hippocampus and neocortex, can synchronize spike timing (*Pouille and Scanziani, 2001*; *Gabernet et al., 2005*) and modulate gain through input normalization (*Pouille et al., 2009*). The PCx is unusual in that it receives its feedforward inhibitory input onto the distal apical dendrites of principal cells, rather than onto perisomatic regions as in other cortices (*Lourenço et al., 2020*). What are the consequences of this atypical architecture for the role of feedforward inhibition in the PCx?

One slice study has shown that feedforward inhibition works together with feedback inhibition to provide spatially and temporally modulated inhibition during bursts of olfactory input (*Stokes and Isaacson, 2010*). Intriguingly, the shift in inhibition reported by *Stokes and Isaacson, 2010*, is in the opposite direction to that found in other brain regions (*Pouille and Scanziani, 2004*; *Silberberg and Markram, 2007*). These results suggest that feedforward inhibition in the PCx is functionally important but has unusual features.

More recently, an in vivo unit recording study in awake mice showed that odor-evoked spiking in feedforward inhibitory neurons in the PCx increases only slowly and weakly after inhalation, suggesting that these neurons provide tonic inhibition and do not play a major role in sculpting odor-evoked responses (*Bolding and Franks, 2018*). Modeling confirmed that feedforward inhibition provides modest subtractive normalization in the PCx, in contrast to the divisive normalization it provides in other cortical regions (*Stern et al., 2018*). An intuitive explanation is that PCx uses a temporal or rate-based code that is more susceptible to 'recurrent normalization' provided by feedback inhibition (*Sturgill and Isaacson, 2015*; *Bolding and Franks, 2018*; *Stern et al., 2018*; *Bolding et al., 2020*; *Pashkovski et al., 2020*). Hence, the evidence of Bolding and colleagues suggests that feedforward inhibition plays a minor role in odor processing under the conditions of their experiments.

The results of Bolding and colleagues seem at variance with our finding that many layer 1a interneurons, especially NG cells, respond vigorously to odors. The discrepancy may lie in the different

recording conditions, for example, whole-cell versus unit recordings, placement of the electrodes (close to or far from the LOT), or our use of anesthesia. Further work is required. However, if feedforward inhibition in the PCx is, indeed, weakly engaged in normal olfactory processing, what other functions might it serve?

The key property of feedforward inhibition in the PCx may be its dendritic localization. By inhibiting the distal apical dendrites of layer 2/3 principal cells, NG and HZ cells have the potential to dampen passive propagation of afferent EPSPs to the soma and to veto dendritic electrogenesis (*Larkum et al., 1999*; *Palmer et al., 2014*; *Pardi et al., 2020*; *Anastasiades et al., 2021*). Indeed, it has been directly shown in vitro that a single layer 1 interneuron can inhibit $Ca^{2+}$ signaling in the distal dendrites of PCx principal cells in a branch-specific fashion (*Stokes et al., 2014*). Feedforward inhibition might then provide a mechanism for regulating processes that involve dendritic electrogenesis and plasticity – including burst-firing (*Tseng and Haberly, 1989*; *Protopapas and Bower, 2001*), spike timing-dependent plasticity (*Kanter et al., 1996*; *Johenning et al., 2009*; *Cassenaer and Laurent, 2012*), and NMDA spikes (*Kumar et al., 2018*; *Kumar et al., 2021*) – and which may only become apparent during experimental paradigms that engage olfactory learning (*Wilson and Stevenson, 2006*; *Ghosh et al., 2015*; *Shakhawat et al., 2015*; *Meissner-Bernard et al., 2019*). Future work would need to explore this possibility.

## Two types of feedforward inhibition

We have previously shown in slices that layer 1a NG cells and HZ cells generate slow-rising (2–3 ms) and fast-rising (1 ms) feedforward unitary IPSPs, respectively, in layer 2 principal neurons, leading us to suggest that NG cell-mediated feedforward inhibition is slower and more diffuse than that provided by HZ cells (*Suzuki and Bekkers, 2012*). In the present report, after taking into account the much slower odor-evoked excitatory drive received by HZ cells, we reach the opposite conclusion. Our results are a reminder that the unitary properties of synapses, measured in vitro, can be less important than the patterns of concerted synaptic activity occurring in vivo.

What might be the value of delivering fast (NG) and slow (HZ) feedforward inhibition? We have shown in a slice simulation (*Figure 7*) that NG cells can strongly inhibit firing in principal cells immediately after a step application of odor. Although this effect was not observed in the study of *Bolding and Franks, 2018*, there nevertheless appears to be scope for a physiological role for this kind of fast inhibition. For instance, NG cells generate GABA transients that spill out of the synaptic cleft and can activate heterosynaptic $GABA_A$ and $GABA_B$ receptors (*Karayannis et al., 2010*; *Overstreet-Wadiche and McBain, 2015*). Future work could explore ways in which this 'volume transmission' could enhance computational complexity in the PCx.

The role of HZ cell-mediated feedforward inhibition is more puzzling. Our slice simulation showed that HZ cells generate a delayed, diffuse response with little effect on principal cell excitability (*Figure 7B*). Trained rodents can discriminate odors within a single sniff (<100 ms; *Uchida and Mainen, 2003*; *Abraham et al., 2004*; *Rinberg et al., 2006*; *Resulaj and Rinberg, 2015*). What could be the relevance of late-firing HZ cells in this scenario? Two unique features of HZ cells suggest that they are potentially important. First, HZ cells are only found close to the LOT (<~200 μm), giving them privileged inhibitory influence over a spatially restricted subset of principal cells (*Suzuki and Bekkers, 2010a*; *Suzuki and Bekkers, 2012*). Second, they are the only class of interneuron in the PCx, and one of few in the cerebral cortex, that are profusely spiny (*Haberly, 1983*). Given the importance of spines for synaptic plasticity (*Engert and Bonhoeffer, 1999*; *Maletic-Savatic et al., 1999*; *Toni et al., 1999*; *Matsuzaki et al., 2004*; *Tonnesen et al., 2014*), HZ cells are poised to receive learning-dependent synaptic excitation, which in turn may allow them to exert adjustable inhibitory control over neurons in their neighborhood.

In summary, we have shown that two forms of odor-evoked feedforward inhibition, fast and slow, are expressed in layer 1a of the PCx. Our work reveals previously unknown circuit elements in the PCx, and adds to a growing understanding of the role of neurons in layer 1 of the cerebral cortex.

# Materials and methods

**Key resources table**

| Reagent type (species) or resource | Designation | Source or reference | Identifiers | Additional information |
|---|---|---|---|---|
| Genetic reagent (*Mus musculus*) | GAD67-GFP (Δneo) | *Tamamaki et al., 2003*, https://doi.org/10.1002/cne.10905 | | GAD67-GFP mice have the EGFP gene targeted to the *Gad1* locus. Breeders were obtained from *Tamamaki et al., 2003*. The colony was maintained on a C57BL6/J background |
| Commercial assay or kit | ABC kit | Vector Laboratories | Vectastain Elite ABC Kit, Cat # PK-6100 | |
| Chemical compound, drug | Urethane | Merck/Sigma-Aldrich | Cat # U2500-100G | 0.7 g/kg s.c. |
| Chemical compound, drug | Chlorprothixene | Merck/Sigma-Aldrich | chlorprothixene hydrochloride, Cat # C1671-1G | 5 mg/kg i.p. |
| Chemical compound, drug | Atropine | Apex Laboratories, Australia | Atropine sulphate, 0.6 mg/ml | 0.2–0.3 mg/kg s.c. |
| Chemical compound, drug | Alexa Fluor 594 | Life Technologies | Alexa Fluor 594 hydrazide, Na salt, Cat # A-10438 | 20 μM |
| Chemical compound, drug | Acetophenone | Merck/Sigma-Aldrich | Cat # A10701-5ML | Flow dilution to 10% of saturated vapor pressure |
| Chemical compound, drug | Anisole | Merck/Sigma-Aldrich | Cat # 96109 | Flow dilution to 10% of saturated vapor pressure |
| Chemical compound, drug | Benzaldehyde | Merck/Sigma-Aldrich | Cat # 418099 | Flow dilution to 10% of saturated vapor pressure |
| Chemical compound, drug | Butyric acid | Merck/Sigma-Aldrich | Cat # 19215–5ML | Flow dilution to 10% of saturated vapor pressure |
| Chemical compound, drug | Ethanol | Merck/Sigma-Aldrich | Cat # 459836 | Flow dilution to 10% of saturated vapor pressure |
| Chemical compound, drug | Ethyl-*n*-butyrate | Merck/Sigma-Aldrich | Cat # 75563 | Flow dilution to 10% of saturated vapor pressure |
| Chemical compound, drug | Eugenol | Merck/Sigma-Aldrich | Cat # 35995 | Flow dilution to 10% of saturated vapor pressure |
| Chemical compound, drug | 1-Heptanal | Merck/Sigma-Aldrich | Cat # 61696 | Flow dilution to 10% of saturated vapor pressure |
| Chemical compound, drug | 2-Heptanone | Merck/Sigma-Aldrich | Cat # 02476 | Flow dilution to 10% of saturated vapor pressure |
| Chemical compound, drug | Lavender oil | Merck/Sigma-Aldrich | Cat # 61718 | Flow dilution to 10% of saturated vapor pressure |
| Chemical compound, drug | Limonene | Merck/Sigma-Aldrich | Cat # 62118 | Flow dilution to 10% of saturated vapor pressure |
| Chemical compound, drug | 1-Pentanol | Merck/Sigma-Aldrich | Cat # 77597 | Flow dilution to 10% of saturated vapor pressure |
| Chemical compound, drug | Propionic acid | Merck/Sigma-Aldrich | Cat # 94425 | Flow dilution to 10% of saturated vapor pressure |
| Chemical compound, drug | Amyl acetate | MP Biomedicals | Cat # 300015 | Flow dilution to 10% of saturated vapor pressure |

*Continued on next page*

*Continued*

| Reagent type (species) or resource | Designation | Source or reference | Identifiers | Additional information |
|---|---|---|---|---|
| Chemical compound, drug | Isoamyl acetate | MP Biomedicals | Cat # 155077 | Flow dilution to 10% of saturated vapor pressure |
| Chemical compound, drug | DNQX | Tocris | 6,7-Dinitroquinoxaline-2,3-dione, Cat # 0189 | 10 µM |
| Chemical compound, drug | D-AP5 | Tocris | D-aminophosphono valeric acid, Cat # 0106 | 50 µM |
| Chemical compound, drug | Picrotoxin | Merck/Sigma-Aldrich | Cat # P1675 | 100 µM |
| Chemical compound, drug | SR 95531 (gabazine) | Hello Bio | Cat # HB0901 | 20 µM |
| Software, algorithm | Igor Pro | Wavemetrics; source code for dynamic clamp procedure is in *Figure 7—source code 1* | https://www.wavemetrics.com/ | Source code folder (*Figure 7—source code 1*) also contains sample data |
| Software, algorithm | Axograph X | Axograph Scientific | https://axograph.com/ | |
| Software, algorithm | Matlab | Mathworks | https://www.mathworks.com/ | |
| Software, algorithm | RStudio | RStudio | https://www.rstudio.com/ | |
| Other | Cal-590 AM | AAT Bioquest | Cat # 20511 (10× 50 µg) | 1 mM (pressure-injected) |

## Animals and surgery for 'in vivo' experiments

All experimental procedures were approved by the Animal Experimentation Ethics Committee of the Australian National University and conform to the Australian Code for the Care and Use of Animals for Scientific Purposes, published by the National Health and Medical Research Council of Australia.

Experiments used heterozygous GAD67-GFP (Δneo) mice, which have the EGFP gene targeted to the *Gad1* locus (*Tamamaki et al., 2003*). The colony was maintained on a C57BL6/J background. Animals were aged 35–75 days and weighed 17–24 g. For surgery, an animal was sedated with chlorprothixene (5 mg/kg i.p.) then injected with urethane (0.7 g/kg s.c.) for general anesthesia plus atropine (0.2–0.3 mg/kg s.c.) to reduce secretions. The correct level of anesthesia was confirmed by observing regular respiration and the complete absence of a pinch reflex; a topup dose of urethane was sometimes required after 5–6 hr. A local anesthetic (prilocaine, 0.2 mg/kg) was applied topically to incision sites during surgery. Access to the PCx was via the cheek and upper mandible, as previously described (*Stettler and Axel, 2009*; *Tantirigama et al., 2017*). Briefly, the following surgical procedure was used (total duration ~2 hr). After retracting the skin, superficial blood vessels of the cheek were cauterized, then the temporalis muscle was carefully detached and retracted toward its base near the eye, revealing the temporal aspect of the skull. The zygomatic bone and the upper sections of the mandible, including the coronoid and the condyloid processes, were removed, followed by the upper section of the masseter muscle, exposing the basolateral surface of the skull. At this point, the PCx was visible under the translucent skull, identifiable using anatomical landmarks such as the LOT and the middle cerebral artery (MCA), which are roughly perpendicular to each other. A metal head post was then glued on top of the skull and the head was stabilized. A craniotomy (~2 mm$^2$) was made with a fine drill (Osada Electric, Nagoya, Japan, or Ram Products Inc, Dayton, NJ) just caudal to the MCA and close to the LOT. This placement of the craniotomy ensured that all recordings were made in the anterior PCx at approximately +0.6 mm from Bregma. In many experiments the dura was left intact, but in some the dura was carefully removed using a needle and fine forceps. After completion

of the surgery, a small chamber made from a plastic weighboat and dental cement was constructed around the site. To keep the area hydrated and allow immersion of the microscope objective, the chamber was filled with a Ringer's solution containing (mM) 135 NaCl, 5.4 KCl, 1.8 $CaCl_2$, 1 $MgCl_2$, 5 HEPES at pH 7.4. For all the above procedures, as well as during recordings, the animal was placed on an electrically heated surface at ~37°C and was kept hydrated by periodic s.c. injections of normal saline with 2% dextrose.

## Two-photon targeted patch clamping

GFP-positive interneurons in GAD67-GFP (Δneo) mice were visualized using a two-photon MOM microscope (Sutter Instrument Company, Novato, CA) with a 40×/0.8 NA water immersion objective (Olympus, Tokyo, Japan) and a Chameleon Ultra Ti:Sapphire laser (Coherent, Santa Clara, CA) tuned to 800–820 nm (*Suzuki et al., 2014*). Frames (512 × 512 pixels) were acquired simultaneously through a red and green channel at ~5 Hz under the control of ScanImage (Vidrio Technologies, Ashburn, VA). Patch pipettes were pulled with a longer taper than usual and had resistances of 5–8 MΩ when filled with internal solution comprising (in mM) 135 $KMeSO_4$, 7 NaCl, 0.1 EGTA, 2 $Na_2ATP$, 2 $MgCl_2$, 0.3 GTP, 10 HEPES at pH 7.2, supplemented with 0.2–0.4% biocytin (295–300 mOs/kg). This solution had a $Cl^-$ concentration of 11 mM and a measured junction potential of –7 mV. For voltage clamp experiments $CsMeSO_3$ replaced $KMeSO_4$. These solutions also contained Alexa Fluor 594 (20 µM) for visualizing the electrode in the red channel. Patch electrodes were positioned using a micromanipulator (MP-285, Sutter Instrument, Novato, CA) and electrical recordings were obtained with a Multi-Clamp 700B amplifier (Molecular Devices, San Jose, CA). Data were filtered at 10 kHz and sampled at 20–50 kHz using an Instrutech ITC-18 digitizing interface (HEKA, Ludwigshafen, Germany) under the control of Axograph X (Axograph Scientific, Sydney, Australia). The reference electrode was a Ag/AgCl wire inserted under the skin. The patch electrode was advanced rapidly to penetrate the dura, then more slowly to approach the selected cell and obtain a gigaseal whole-cell recording in the usual way (*Margrie et al., 2002*). For current clamp recordings, bridge balance and capacitance neutralization were adjusted and the cell was allowed to remain at its resting potential. For voltage clamp recordings, series resistance compensation was not used. Cells were included in the dataset if they had a mean resting potential more hyperpolarized than –50 mV and were stable enough to allow the recording of responses to at least five odors. In addition, cells had to be unambiguously identified as either NG or HZ cells according to the criteria given in the Results. At the end of the recording an image stack of the cell was acquired in both the red and green channels.

## Blind 'in vivo' patch clamping

This method was used to measure the EPSGs used in the in vitro simulation in *Figure 7B*. The dura was removed but a coverslip was not used. Patch electrodes were prepared and filled as for targeted patch clamping, then an electrode was positioned at the surface of the anterior PCx using a dissection microscope. The micromanipulator (MP-285, Sutter Instrument) was zeroed at the surface, then the electrode was advanced rapidly to the search depth (150–280 µm, corresponding to layer 2) while applying high pressure (25 kPa), after which the pressure was reduced (4–8 kPa) and a whole-cell recording obtained using standard techniques (*Margrie et al., 2002*; *Poo and Isaacson, 2009*). Data were acquired as described above. EPSGs were calculated from EPSCs recorded under voltage clamp at a holding potential of –70 mV, close to the chloride reversal potential for these solutions. Layer 2 principal neurons (SL and SP cells) were identified by their intrinsic electrical properties (*Suzuki and Bekkers, 2006*) and by the recording depth (SL: 150–200 µm; SP: 200–280 µm). Cell identity was also confirmed by fixing the brain at the end of the experiment and recovering the morphology of the recorded neuron as previously described (*Suzuki et al., 2014*).

## 'In vivo' functional Ca²⁺ imaging

Imaging used the red-shifted $Ca^{2+}$ indicator Cal-590 AM (AAT Bioquest, Sunnyvale, CA), which was prepared and injected as previously described (*Tischbirek et al., 2015*). Briefly, the dura was removed and dye (1 mM) was pressure-injected into the PCx at a depth of ~200 µm using a glass pipette (tip diameter ~10 µm). A coverslip was glued over the PCx and imaging commenced >1 hr after injection. Imaging frames were acquired at 30 Hz using a custom-modified B-scope two-photon microscope (Thorlabs, Newton, NJ) with a 16×/0.8 NA water immersion objective (Nikon, Tokyo, Japan),

resonance-galvanometer scanners and a Chameleon Ultra Ti:Sapphire laser (Coherent, Santa Clara, CA) tuned to 800–820 nm. Cells were included in the dataset if they unambiguously satisfied the fluorescence and soma morphology criteria given in Results, and if they exhibited clear odor responses according to the criteria under Data analysis, below.

## Odor presentation

A custom-built flow-dilution olfactometer was used to deliver up to 15 odors which were diluted to 10% of their saturated vapor pressure in charcoal-filtered medical air (flow rate 1 L/min; *Bozza et al., 2004*). The odors used in this study were: acetophenone, anisole, benzaldehyde, butyric acid, ethanol, ethyl-*n*-butyrate, eugenol, 1-heptanal, 2-heptanone, lavender oil, limonene, 1-pentanol, propionic acid (Merck/Sigma-Aldrich, St Louis, MO), amyl acetate and isoamyl acetate (MP Biomedicals, Sydney, Australia). Odors were presented for 3 s at 60 s intervals. Control experiments used a miniPID photo-ionization device (Aurora Scientific, Aurora, Canada) to confirm that odors were presented in a step-like manner (20–80% rise time in 46 ± 2 ms, *n* = 9 odors; *Figure 3—figure supplement 1*). However, there was a consistent delay of 242 ± 6 ms (*n* = 9) between the switching time of the final valve and the arrival of the odorant at the detector inlet (*Figure 3—figure supplement 1*). All odor arrival times in the paper have been corrected for this delay. The mouse was freely breathing and its respiration was recorded using a piezoelectric strap (Kent Scientific, Torrington, CT) around the abdomen. Control experiments confirmed that the onset of inhalation coincided with the start of the downward step visible in some respiration traces (e.g. *Figure 1—figure supplement 1*) and the beginning of exhalation corresponded to the peak of the upward spike. Because the upward spike was a more reliable feature, we estimated inhalation onset by reference to the exhalation. In a subset of experiments in which both features were clear, we measured the mean latency from the peak of the exhalation spike to the start of the following inhalation (239 ± 8 ms; mean time between exhalation spikes, 365 ± 12 ms; *n* = 10 mice). Hence, expressed as a fraction of a respiration cycle, inhalation onset occurred at 0.654 ± 0.007 after the exhalation spike; this was used to estimate the time of onset of inhalation for all experiments. Odor stimulus onset was defined as the time of the first estimated onset of inhalation that occurred following the corrected odor arrival time. To avoid habituation, each odor was presented only once. Cross-habituation between different odors was not observed.

## Slice experiments

Coronal slices (300 µm thick) were obtained from the anterior PCx of GAD67-GFP (Δneo) mice aged 20–30 days, as previously described (*Suzuki and Bekkers, 2006*; *Suzuki and Bekkers, 2012*). Briefly, slices were prepared on a tissue slicer (Campden Instruments, Loughborough, UK) in ice-cold high-Mg$^{2+}$ cutting solution comprising (in mM) 125 NaCl, 3 KCl, 0.5 CaCl$_2$, 6 MgCl$_2$, 25 NaHCO$_3$, 1.25 NaH$_2$PO$_4$, 2 ascorbate, 3 pyruvate, and 10 glucose (osmolarity 305 mOs/kg), bubbled with 5% CO$_2$/95% O$_2$ (carbogen). The slices were incubated for 40 min at 34°C in carbogen-bubbled artificial cerebrospinal fluid (ACSF; composition below) then were held at room temperature until required.

Whole-cell patch-clamp recordings were made from visually identified GFP-positive interneurons using an Olympus BX51WI microscope equipped with infrared differential interference contrast and wide-field fluorescence, as described previously (*Suzuki and Bekkers, 2012*). Slices were superfused with warmed ACSF containing (in mM) 125 NaCl, 3 KCl, 2 CaCl$_2$, 1 MgCl$_2$, 25 NaHCO$_3$, 1.25 NaH$_2$PO$_4$, and 25 glucose (310 mOs/kg), bubbled with carbogen and maintained at 33°C ± 1°C. For the experiments in *Figures 6B and 7* the bath solution contained 6,7-dinitroquinoxaline-2,3-dione (DNQX, 10 µM) and D-aminophosphonovaleric acid (D-AP5, 50 µM) to block ionotropic glutamate receptors (Tocris, Abingdon, UK). For the experiments in *Figure 5B* the bath solution instead contained 100 µM picrotoxin (Merck/Sigma-Aldrich), while in *Figure 6C* and *Figure 6—figure supplement 2* it contained 20 µM SR 95531 (gabazine; HelloBio, Bristol, UK), in both cases to block GABA$_A$ receptors. Patch electrodes had resistances of 4–6 MΩ when filled with KMeSO$_4$- or CsMeSO$_3$-based internal solution (same as used in vivo). Unless stated otherwise, compounds were obtained from Merck/Sigma-Aldrich.

Recordings were obtained using the same instrumentation and software as for the in vivo experiments. The stimulating electrode was made from a low-resistance patch electrode (~1 MΩ) filled with ACSF and coated with conductive paint. For connected pair recordings (*Figure 6B*), the pre- and post-synaptic electrodes were filled with KMeSO$_4$- and CsMeSO$_3$-based internal solutions, respectively,

allowing the postsynaptic neuron to be voltage clamped at a holding potential of +50 mV while the presynaptic neuron was allowed to remain at its resting potential in current clamp mode. For the dual recordings (*Figure 6C*, *Figure 6—figure supplement 2*), both recording electrodes were filled with the KMeSO$_4$-based internal solution, and trains of extracellular stimuli (5 × 200 μs at 20 Hz) were delivered by a constant-voltage stimulator (Digitimer, Welwyn Garden City, UK) over the range 0–85 V in 0.1–5 V increments. Cells were included in the dataset if they had a stable resting potential more hyperpolarized than –60 mV (current clamp) or a stable holding current <100 pA when held at –70 mV (voltage clamp). At the end of the experiment the slice was fixed and processed with an ABC kit (Vector Laboratories, Burlingame, CA), allowing recovery of the morphologies of the recorded neurons.

## Patterned stimulation and dynamic clamp experiments in slices

In vivo-like patterns of inhibitory synaptic stimulation in slice experiments (*Figure 7*) were adjusted to a common respiration time base as follows. First, a subset of odor-evoked firing patterns from NG and HZ cells, together with their associated respiration traces, was randomly selected from the full dataset for each cell type (sample data in *Figure 7—source code 1*). For each subset, one firing/respiration combination was chosen as a reference, and the first upward peak in the reference respiration trace after odor onset was defined as $t_0$. Every other odor-evoked firing pattern in that subset was translated in time to align its first respiration peak after odor onset to $t_0$. Working forward and backward from $t_0$, for each respiration interval the respiration trace (and associated firing pattern) of each other firing pattern was excised from the original data recording and linearly stretched ('warped') so the duration of that respiration interval matched the corresponding interval in the reference respiration trace. Finally, all of these warped segments were concatenated in their original order, and the resultant stimulus patterns (examples in *Figure 7A*, red traces) were used as the trigger to the extracellular stimulator in slice experiments. The same method was used to align the EPSG to the inhibitory stimulation patterns (*Figure 7B*, *Figure 7—source code 1*). In this case the respiration trace for the EPSG was used as the reference time base.

The EPSG used in *Figure 7B* was obtained from an EPSC recorded blind in vivo from an SP cell in response to a 3-s-long application of ethyl-*n*-butyrate (data in *Figure 7—source code 1*). Although the in vivo EPSC was measured near the chloride reversal potential to minimize contamination due to direct inhibition onto that cell, the EPSC will still be the result of excitatory input from all other neurons in the circuit, including other SP cells that have received their own feedforward inhibition. Furthermore, injecting an excitatory conductance at the soma does not accurately replicate excitatory synaptic inputs distributed across the dendrites. These details were disregarded for the purpose of this simple in vitro simulation. For each neuron the conductance magnitude was adjusted to produce a similar firing rate with the extracellular stimulus switched off, then the stimulator was switched on to record the effect of synaptic inhibition. To provide a reference firing rate for normalizing the PSTH (*Figure 7B*), a fixed conductance stimulus was inserted at the end of the EPSG, well past the odor period (not visible in *Figure 7B*). The dynamic clamp was implemented using Igor Pro (Wavemetrics, Lake Oswego, OR; *Figure 7—source code 1*).

## Data analysis

All analysis was done using Igor Pro, Axograph X, Matlab (MathWorks, Natick, MA), or R (running under RStudio, Boston, MA).

AP properties (*Figure 1D*) were measured as previously described for slice experiments (*Suzuki and Bekkers, 2010a*). Latency to the first AP was the delay from the beginning of the current step to the first AP at rheobase. AP height was the difference between the peak of the first AP at rheobase and its threshold voltage (defined as the membrane potential [$V_m$] at which $dV_m/dt$ first exceeded 15 V/s). AP halfwidth was the width of the first AP at rheobase measured at half its height. Respiration-correlated oscillations in $V_m$ (*Figure 2*) were characterized as follows. Respiration traces were band-pass filtered at 1–50 Hz and normalized to oscillate between 0 and 1. The time between successive exhalation spikes was used to calculate the mean respiration frequency in three 3-s-long windows (–6 to –3 s, 0–3 s, and 16–19 s, all with respect to odor onset). The same three windows were used for the analysis of oscillations in $V_m$. First, $V_m$ was median-filtered to remove APs. Mean peak-to-peak amplitude of the oscillations in $V_m$ was found by excising segments of $V_m$ between successive exhalation spikes, linearly

stretching ('warping') them so they had the same time axis, then averaging together all segments within each of the three analysis windows. Mean peak-to-peak amplitude was taken as the difference between the maximum and minimum amplitude of the averaged segment of $V_m$. For cross-covariance analysis (*Figure 2—figure supplement 1*, *Figure 2—figure supplement 2*), the respiration trace was replaced by a normalized series of Gaussians representing the upward spikes in the respiration cycle; this was done to eliminate the effect of differences in the amplitude of the recorded respiration trace. The amplitude of the largest positive peak in the covariance was found, as well as the location of this peak expressed as a fraction of the mean time between exhalation spikes for that window.

The odor response indices were found by: (i) calculating the fraction of odors that each cell responded to, then averaging across cells ('cell-averaged index') and (ii) calculating the fraction of cells each odor activated, then averaging across odors ('odor-averaged index'). Odor responsiveness was determined using a z-score criterion: median-filtered $V_m$ segments demarcated by exhalation spikes were excised as above, and the mean $V_m$ in each of these segments was calculated for the entire odor trial. These values were converted to a z-score by subtracting the mean and dividing by the standard deviation of the list of mean $V_m$ values during the 8-s-long pre-odor baseline period. An odor response was said to occur if this z-score value exceeded 2.5 for any respiration segment during the 3-s-long odor application period (*Figure 1—figure supplement 1*). Lifetime and population sparseness were calculated with modified expressions (*Pashkovski et al., 2020*) that used the peak z-scored response (without thresholding) as the variable.

Individual odor-evoked EPSPs (*Figure 4A*), EPSCs (*Figure 5C*), and IPSCs (*Figure 6A*) were notch-filtered at 2–4 Hz, if required, to remove respiration-associated oscillations prior to averaging (see also *Figure 4—figure supplement 1*). To estimate the latency to peak and halfwidth, each individual EPSC was fitted to the equation $I(t) = a(1 - e^{-t/\tau_1})^2 e^{-t/\tau_2}$, where $a$ is amplitude and $\tau_1$ and $\tau_2$ are the rising and falling time constants, respectively. The latency and halfwidth were then measured from this fitted curve.

In vivo $Ca^{2+}$ imaging experiments (*Figure 3*, *Figure 3—figure supplement 2*) were analyzed by manually drawing regions of interest around the somas of identified neurons in the green channel then measuring mean somatic fluorescence, $F$, in the red channel for each frame. Baseline fluorescence ($F_0$) for each neuron was defined as the median of the lower 80% of all values of $F$ measured in all frames acquired prior to odor application. Plots show $\Delta F/F_0 = (F - F_0)/F_0$.

Connected-pair recordings (*Figure 6B*) and dual recordings (*Figure 6C*) in slices were analyzed as follows. The peak postsynaptic current was found within a 5- or 10-ms-long window starting 2 ms after the time of the presynaptic AP or extracellular stimulus, then the mean peak amplitude ($a_{pk}$) was found by averaging over a 1 ms long window around the peak and subtracting the averaged baseline over a 5- or 10-ms-long window ending 2 ms before the time of the presynaptic AP or extracellular stimulus. For the connected-pair recordings (*Figure 6B*) the same measurement procedure was repeated for 200 randomly-chosen times during the 500-ms-long baseline period preceding the time of the presynaptic AP, and the standard deviation ($s_b$) of these baseline mean amplitudes was calculated. A synaptic connection was identified if $|a_{pk}| > |3 s_b|$. For the dual recordings (*Figure 6C*) the mean NG:HZ EPSC amplitude ratio was calculated as follows. For each stimulator setting, all evoked EPSC amplitudes at that setting (typically $n = 5$) were averaged together and the ratio was calculated. In the case of experiments with a clear plateau (e.g. *Figure 6C*, bottom left), the individual ratios for EPSCs on the plateau were averaged together to yield the overall mean NG:HZ ratio for that experiment. In the case of experiments without a clear plateau (e.g. *Figure 6—figure supplement 2*), the overall mean only included individual ratios for which both EPSCs were >10 pA.

## Statistical analysis

All statistical analysis was done using R (version 3.6.0) running under RStudio. Sample sizes were not predetermined using a statistical test; we established that our sample sizes were sufficient from the size and statistical significance of the results, and our sizes are similar to those commonly used in the field. Data collection and analysis were not blinded or randomized, but analysis was automated whenever possible. Results are presented as mean ± standard error of the mean (SEM) with associated exact p value ($n$ = number of cells, cell pairs, or cell-odor pairs, as indicated). Pairwise comparisons were done using Welch's unpaired two-tailed t-test (*t.test* function in R). Multiple paired comparisons used one-way ANOVA with Tukey's contrasts (*lmer* and *glht* functions in R) with blocking by cell and

odor number. Visual inspection of residual plots did not reveal any large deviations from homoscedasticity. Distributions were compared using the KS test (*ks.test*). Significance is indicated on the figures by *ns* (not significant) or by one or more asterisks, following the usual convention in R.

## Additional information

### Funding

| Funder | Grant reference number | Author |
|---|---|---|
| National Health and Medical Research Council | 1009382 | John M Bekkers |
| National Health and Medical Research Council | 1050832 | John M Bekkers |

The funders had no role in study design, data collection and interpretation, or the decision to submit the work for publication.

### Author contributions

Norimitsu Suzuki, Malinda LS Tantirigama, Conceptualization, Data curation, Formal analysis, Investigation, Methodology, Software, Validation, Visualization, Writing – review and editing; K Phyu Aung, Data curation, Investigation, Methodology, Writing – review and editing; Helena HY Huang, Data curation, Investigation, Methodology, Validation; John M Bekkers, Conceptualization, Data curation, Formal analysis, Funding acquisition, Investigation, Methodology, Project administration, Resources, Software, Supervision, Validation, Visualization, Writing – original draft, Writing – review and editing

### Author ORCIDs

Norimitsu Suzuki (iD) http://orcid.org/0000-0002-9338-9099
Malinda LS Tantirigama (iD) http://orcid.org/0000-0003-0791-9389
K Phyu Aung (iD) http://orcid.org/0000-0003-3744-5321
Helena HY Huang (iD) http://orcid.org/0000-0002-1283-8586
John M Bekkers (iD) http://orcid.org/0000-0001-8619-5512

### Ethics

This work was performed in strict accordance with the guidelines contained in the Australian Code for the Care and Use of Animals for Scientific Purposes, published by the National Health and Medical Research Council of Australia. All experimental procedures were approved by the Animal Experimentation Ethics Committee of the Australian National University (Protocols A2014/11, A2017/27, A2018/52). All surgeries and experiments were performed under anesthesia with either urethane or fentanyl/medetomidine, with supplementary isoflurane as required, and every effort was made to minimize suffering.

### Decision letter and Author response

Decision letter https://doi.org/10.7554/eLife.73406.sa1
Author response https://doi.org/10.7554/eLife.73406.sa2

## Additional files

### Supplementary files

• Transparent reporting form

### Data availability

Source data files have been provided for Figures 1, 2 and 4. Source code and sample data have been provided for Figure 7.

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
