## [Editor Report]

Feedforward inhibition (FFI) typically exerts a powerful effect shaping neural activity. In this paper, Suzuki et al., use a combination of in vivo and in vitro experiments to characterize, for the first time, responses in the two main classes of FFIs in the mouse olfactory cortex, neurogliaform cells (NG) and horizontal cells (HZ). They find that these two cell types have different responses and different connectivity, which partially explains their different responses. This paper also helps resolve a previously perplexing result from a recent publication proposing that FFI in the mouse olfactory cortex plays a negligible role in shaping cortical odor responses, presumably because those authors were only recording from HZ, but not NG, cells.

---

## [Decision Letter]

**Decision letter after peer review:**

Thank you for submitting your article "Multi-functional feedforward inhibitory circuits for cortical odor processing" for consideration by *eLife*. Your article has been reviewed by 3 peer reviewers, and the evaluation has been overseen by a Reviewing Editor and John Huguenard as the Senior Editor. The following individual involved in review of your submission has agreed to reveal their identity: J. Julius Zhu (Reviewer #3).

Essential revisions:

1) As Reviewer #1 suggested, the experiments shown in Figure 5 need to be redone with inhibition blocked.

2) Please pay attention to the different inputs to HZ and NG. Reviewer #1 suggests that you can address this relatively straightforwardly by doing paired recordings from HZ and NG cells and stimulating the LOT through a range of strengths, including minimal stimulation. E.g. Stokes and Isaacson did this for a FFI and PYR and showed they got inputs from the same LOT fiber. Such careful characterization can strengthen the conclusion and improve the manuscript.

3) There are discrepancies on the strengths of inhibitory inputs onto HZ and NG cells. For example, the manuscript nicely shows strong, almost equivalent inhibition in HZ and NG cells in vivo, but the connectivity analyses show almost no inhibitory inputs onto NGs. So who is inhibiting them? Additional experiments need to be performed to address this issue.

4) Both Reviewer #1 and #2 feel that the functional implications (as stated in the title) are lacking, please tune down conclusions here.

5) Please add the missing quantification, details requested by the reviewers.

*Reviewer #1 (Recommendations for the authors):*

– I also don't understand the warping of the respiration traces. Weren't the authors just stimulation with defined electrical patterns? Were these variable, then "warped" to make them the same? Along these lines, I don't think the respiration trace should be included here. If it is the authors need to explain where it comes from.

L. 94. Franks et al., 2011 did not record in vivo.

L. 344-5 – “Our findings reveal a remarkable richness in inhibitory control at the very first stage of cortical odor processing…” Their findings show differences in the responses they recorded in NG vs. HZ cells in slices or to strong and prolonged odor pulses in deeply anesthetized mice that are missing half their face without any perturbation or analysis of functional significance. Let’s tone down the hyperbole a bit.

L. 365-6 – The Fontanini and Tantirigama citations are correct but the takeaway from Bolding et al., 2020 was that the piriform cortex was surprisingly robust to ketamine/xylazine anesthesia,

l. 477-83. The authors should also cite the recent paper from Jackie Schiller's group (Amit et al., 2018) that shows that piriform neurons can support NMDA spikes. Dendritic-targeting inhibition may be important for gating these too.

*Reviewer #2 (Recommendations for the authors):*

The in vivo experiments performed in this study are incredibly difficult and the findings are fundamental to understanding the circuitry that underlies olfactory processing. This novel characterization of the odor response properties of NG and HZ interneurons is well done. The main finding that NG and HZ cells broadly tuned but show differing responses in strength and temporal dynamics is compelling. It is also interesting that the underlying excitatory synaptic dynamics in part, contribute to differences in odor response properties. The authors addressed a discrepancy between the short-term plasticity of afferent excitatory synaptic inputs recorded in vitro and in vivo responses. Although this only partially explained responses in NG neurons but not HZ neurons. The authors also provide insight into inhibitory dynamics between NG and HZ neurons that also might contribute to the overall response properties of these classes. Finally, the authors use dynamic clamp to investigate a model of how the NG and HZ interneurons influence the feedforward processing of superficial pyramidal neurons. This last set of experiments was underwhelming. Overall this study provides a solid characterization of the odor response properties of NG and HZ interneurons. The study inspires several future directions. However, the mechanisms by which these two classes are integrated into the circuit and function in feedforward odor processing are less well developed and makes the study seem incomplete.

1) Both NG and HZ neurons show comparable broad tuning for odor identity. Was there any attempt to investigate the concentration dependence of their firing rates? Recent studies from the Rinberg and Schaefer labs have shown that putative tufted cells show concentration invariant onset responses possibly similar to NG neurons while mitral cells show more prolonged responses like Hz neurons that may be concentration dependent. Is there a parallel here? The authors do raise the possibility that M/T cells might differentially excite NG and HZ cells resulting in the different time courses of the excitatory synaptic inputs. Investigating the concentration dependence of odor responses could provide some insight.

2) In the absence of different excitatory drive to NG or HZ neurons are there potentially differences in dendritic filtering that could underlie the fast vs slow integration of excitation.

3) The difficulties of assessing the functional roles of the two classes interneurons in vivo are appreciated, so using dynamic clamp (or computational modeling) is a valid option. But the dynamic clamp experiments seem contrived. It is not surprising that strong onset firing modeled for stimulation of NG neurons evokes strong onset inhibition in pyramidal neurons that in turn decreases onset firing rates. Moreover, this tool provided no insight to how HZ neurons transform pyramidal neuron responses. However, dynamic clamp might be useful in an inverse approach. The EPSC recorded in vivo and used to generate the EPSG is likely not purely excitatory and is probably already influenced by NG or HZ inputs (as are pyramidal neuron spike recordings in vivo). Could the information attained in the present study about the spiking patterns and synaptic inhibition (previous work from the lab) be used to design inhibitory conductances that could effectively be subtracted from overall conductances based on in vivo recordings of pyramidal neurons? This might mirror an optogenetic silencing study. If this were to be applied to an array of pyramidal neurons odor responses that vary by timing, tuning or concentration dependence one could get some insight into how these might have been shaped by feedforward inhibition from NG or HZ neurons in isolation or conjunction. This could be particularly interesting with respect to responses from neurons that are suppressed by odor inputs, or are weakly responsive as have also been shown by this group.

*Reviewer #3 (Recommendations for the authors):*

It seems that the missed the opportunity to examine the odor-mimic stimulation-evoked EPSPs and EPSCs when performing the simultaneous recordings from both L1 neurogliaform and horizontal cells. These experiments would address the hypothesis that the heterogeneous lateral olfactory tract fiber tracts may differentially innervate two types of L1 interneurons. The results should also help to reveal the kinetics of evoked responses, validate the exact effect of "distant" NG stimulus and "near" HZ stimulus on these two cell types, and later use the observed responding EPSPs to test their impacts on postsynaptic pyramidal neurons. The impact of this study would be significantly improved if the authors choose to include such experiments.

---

## [Author Response]

Essential revisions:1) As Reviewer #1 suggested, the experiments shown in Figure 5 need to be redone with inhibition blocked.

These experiments in fact included 100 µM picrotoxin in the bath, as we routinely do for experiments of this kind (see *e.g.* Suzuki and Bekkers, 2006, 2010, 2011). We apologize for forgetting to mention this. The information has now been added to the Results (p 12), Materials and methods (pp 35-36) and Figure 5 legend.

2) Please pay attention to the different inputs to HZ and NG. Reviewer #1 suggests that you can address this relatively straightforwardly by doing paired recordings from HZ and NG cells and stimulating the LOT through a range of strengths, including minimal stimulation. E.g. Stokes and Isaacson did this for a FFI and PYR and showed they got inputs from the same LOT fiber. Such careful characterization can strengthen the conclusion and improve the manuscript.

We have now performed these paired-recording experiments and the results are shown in the new Figure 6C and Figure 6 —figure supplement 2. In brief, we found that, in response to a train of stimuli applied to the LOT, the first EPSC in the train was always smaller in the NG cell than in the HZ cell. For later EPSCs in the train, however, the amplitudes in the two cell types became more similar because of the previously-reported differences in short-term plasticity at these two inputs (Suzuki and Bekkers, 2010; see also upper right panel in Figure 6C). This finding is now presented in the Results (pp 14-15) and Discussion (p 22). These new experiments were performed by K. Phyu Aung, who has been added to the author list.

3) There are discrepancies on the strengths of inhibitory inputs onto HZ and NG cells. For example, the manuscript nicely shows strong, almost equivalent inhibition in HZ and NG cells in vivo, but the connectivity analyses show almost no inhibitory inputs onto NGs. So who is inhibiting them? Additional experiments need to be performed to address this issue.

We think the NG cells are inhibited by other NG cells (Results, p 14). This is consistent with the connectivity analysis (Figure 6B) which shows about 12% NG→NG cell connectivity. In fact, this NG→NG connectivity is not significantly different from NG→HZ connectivity (*p* = 0.1, Chi-square 2 x 2 contingency test) although the *n*-values are modest. We unwisely downplayed the NG→NG connectivity in the previous version of the manuscript by calling it “comparatively rare”. This has been rephrased (p 14) and further details have been added to Figure 6B and its legend.

4) Both Reviewer #1 and #2 feel that the functional implications (as stated in the title) are lacking, please tune down conclusions here.

A number of changes have been made to reduce the emphasis on the functional implications, including:

– The title has been changed from “Multi-functional feedforward inhibitory…” to “Fast and slow feedforward inhibitory…” to stress the observations rather than the function.

– In the Introduction (p 5), “Thus, a multi-functional feedforward inhibitory circuit exists…” has been changed to “Thus, two distinctive feedforward inhibitory circuits exist…”.

– In the Discussion (p 18), “Our findings reveal a remarkable richness in inhibitory control at the very first stage of cortical odor processing…” has been changed to “Our findings reveal different types of inhibitory responses at the first stage of cortical odor processing…”.

– Also in the Discussion (p 26), “Our work reveals a previously unknown multi-functional circuit element…” has been changed to “Our work reveals previously unknown circuit elements…”.

5) Please add the missing quantification, details requested by the reviewers.

Please see below for our detailed responses.

Reviewer #1 (Recommendations for the authors):– I also don't understand the warping of the respiration traces. Weren't the authors just stimulation with defined electrical patterns? Were these variable, then "warped" to make them the same? Along these lines, I don't think the respiration trace should be included here. If it is the authors need to explain where it comes from.

Yes, the in vivo-recorded odor responses were all on different time bases (where ‘time base’ means the respiration pattern) and we wanted to make them the same. The purpose of warping was to align the odor responses that were recorded on different trials to a common time base, *i.e.* respiration pattern. This allowed us to average across different stimulus patterns without averaging out respiration-synchronized oscillations of the kind shown in Figure 2. We want to show the reference respiration trace (Figure 7A, bottom) because this allows the reader to see that the oscillations in the averaged IPSPs (Figure 7A, second from bottom) are synchronized with the respiration. We have tried to clarify this by making changes in the Results (p 16) and Materials and methods (p 37). We also now provide a folder (Figure 7 – source code 1) which contains the Igor source code, sample data, and a ReadMe file giving details on how to install and run the code.

L. 94. Franks et al., 2011 did not record in vivo.

This has been corrected.

L. 344-5 – “Our findings reveal a remarkable richness in inhibitory control at the very first stage of cortical odor processing…” Their findings show differences in the responses they recorded in NG vs. HZ cells in slices or to strong and prolonged odor pulses in deeply anesthetized mice that are missing half their face without any perturbation or analysis of functional significance.

We have changed this to read, “Our findings reveal different types of inhibitory control at the first stage of cortical odor processing…” (Discussion, p 18).

L. 365-6 – The Fontanini and Tantirigama citations are correct but the takeaway from Bolding et al., 2020 was that the piriform cortex was surprisingly robust to ketamine/xylazine anesthesia,

This has been corrected.

l. 477-83. The authors should also cite the recent paper from Jackie Schiller's group (Amit et al., 2018) that shows that piriform neurons can support NMDA spikes. Dendritic-targeting inhibition may be important for gating these too.

This reference has been added, as well as another from Jackie Schiller’s group that has recently been published (Kumar *et al.*, 2021) (Discussion, p 24).

Reviewer #2 (Recommendations for the authors):The in vivo experiments performed in this study are incredibly difficult and the findings are fundamental to understanding the circuitry that underlies olfactory processing. This novel characterization of the odor response properties of NG and HZ interneurons is well done. The main finding that NG and HZ cells broadly tuned but show differing responses in strength and temporal dynamics is compelling. It is also interesting that the underlying excitatory synaptic dynamics in part, contribute to differences in odor response properties. The authors addressed a discrepancy between the short-term plasticity of afferent excitatory synaptic inputs recorded in vitro and in vivo responses. Although this only partially explained responses in NG neurons but not HZ neurons. The authors also provide insight into inhibitory dynamics between NG and HZ neurons that also might contribute to the overall response properties of these classes. Finally, the authors use dynamic clamp to investigate a model of how the NG and HZ interneurons influence the feedforward processing of superficial pyramidal neurons. This last set of experiments was underwhelming. Overall this study provides a solid characterization of the odor response properties of NG and HZ interneurons. The study inspires several future directions. However, the mechanisms by which these two classes are integrated into the circuit and function in feedforward odor processing are less well developed and makes the study seem incomplete.1) Both NG and HZ neurons show comparable broad tuning for odor identity. Was there any attempt to investigate the concentration dependence of their firing rates? Recent studies from the Rinberg and Schaefer labs have shown that putative tufted cells show concentration invariant onset responses possibly similar to NG neurons while mitral cells show more prolonged responses like Hz neurons that may be concentration dependent. Is there a parallel here? The authors do raise the possibility that M/T cells might differentially excite NG and HZ cells resulting in the different time courses of the excitatory synaptic inputs. Investigating the concentration dependence of odor responses could provide some insight.

This is a good suggestion, but unfortunately we did not look at the effect of changing odor concentration.

2) In the absence of different excitatory drive to NG or HZ neurons are there potentially differences in dendritic filtering that could underlie the fast vs slow integration of excitation.

We don’t think dendritic filtering alone is able to explain the difference. As mentioned in the Results (p 12) and reported previously (Suzuki and Bekkers, 2010, 2012), the rise time of electrically-evoked individual EPSPs in NG and HZ cells (milliseconds) is much faster than the rise time of odor-evoked compound EPSPs reported here (hundreds of milliseconds). Differences in dendritic filtering could only account for millisecond differences. For this reason we disregarded dendritic filtering and instead focused on how slowly-rising compound EPSPs might be constructed from trains of individual EPSPs (Figure 5A, B). This has now been clarified (Results, p 12).

3) The difficulties of assessing the functional roles of the two classes interneurons in vivo are appreciated, so using dynamic clamp (or computational modeling) is a valid option. But the dynamic clamp experiments seem contrived. It is not surprising that strong onset firing modeled for stimulation of NG neurons evokes strong onset inhibition in pyramidal neurons that in turn decreases onset firing rates. Moreover, this tool provided no insight to how HZ neurons transform pyramidal neuron responses. However, dynamic clamp might be useful in an inverse approach. The EPSC recorded in vivo and used to generate the EPSG is likely not purely excitatory and is probably already influenced by NG or HZ inputs (as are pyramidal neuron spike recordings in vivo).

We agree that the dynamic clamp approach is not optimal but, as we say in the Results (p 16), it is difficult to perform a ‘functional’ experiment like this in vivo. The EPSC used to generate the EPSG was recorded near the chloride reversal potential to minimize contamination due to direct inhibition onto that cell. However, it is true that the EPSC will still be the result of excitatory input from all other neurons in the circuit, including other SP cells that have received their own feedforward inhibition. This and other limitations of our dynamic clamp approach are now made more explicit in the Materials and methods (pp 37-38).

Could the information attained in the present study about the spiking patterns and synaptic inhibition (previous work from the lab) be used to design inhibitory conductances that could effectively be subtracted from overall conductances based on in vivo recordings of pyramidal neurons? This might mirror an optogenetic silencing study. If this were to be applied to an array of pyramidal neurons odor responses that vary by timing, tuning or concentration dependence one could get some insight into how these might have been shaped by feedforward inhibition from NG or HZ neurons in isolation or conjunction. This could be particularly interesting with respect to responses from neurons that are suppressed by odor inputs, or are weakly responsive as have also been shown by this group.

This is an interesting suggestion but we feel there are too many ill-defined parameters to make it feasible at this time. An optogenetic or chemogenetic silencing study would be the way to go in the future, we believe, as soon as techniques are available for selective expression of transgenes in layer 1a NG and HZ cells.

Reviewer #3 (Recommendations for the authors):It seems that the missed the opportunity to examine the odor-mimic stimulation-evoked EPSPs and EPSCs when performing the simultaneous recordings from both L1 neurogliaform and horizontal cells. These experiments would address the hypothesis that the heterogeneous lateral olfactory tract fiber tracts may differentially innervate two types of L1 interneurons. The results should also help to reveal the kinetics of evoked responses, validate the exact effect of "distant" NG stimulus and "near" HZ stimulus on these two cell types, and later use the observed responding EPSPs to test their impacts on postsynaptic pyramidal neurons. The impact of this study would be significantly improved if the authors choose to include such experiments.

We thank the reviewer for this suggestion. We have done new experiments in which we made simultaneous whole-cell recordings from NG and HZ cells while applying both minimal and strong electrical stimulation to the LOT. The results are shown in the new Figure 6C and Figure 6 —figure supplement 2. Briefly, we found that, when applying a train of stimuli to the LOT, the first EPSC was always smaller in the NG cell than in the HZ cell, even when using ‘minimal’ stimulation or when the NG cell was closer to the LOT. After the second or third EPSC in the train, however, the EPSC amplitudes in the two cell types became more similar because of the previously-reported differences in short-term plasticity at these two inputs (Suzuki and Bekkers, 2010; see also upper right panel in Figure 6C). Given the dynamics of afferent input from the olfactory bulb (*i.e.* modulated bursts of APs), these results suggest that NG and HZ cells mostly operate in a regime where they receive similar afferent excitation. This finding is now discussed in the Results (pp 14-15) and Discussion (p 22).